# Towards Anomaly-Aware Pre-Training and Fine-Tuning for Graph Anomaly Detection

**Yunhui Liu**[1]*, **Jiashun Cheng**[2]*, **Yiqing Lin**[3], **Qizhuo Xie**[1],
**Jia Li**[2], **Fugee Tsung**[2], **Hongzhi Yin**[4], **Tao Zheng**[1], **Jianhua Zhao**[1], **Tieke He**[1]†
[1] State Key Laboratory for Novel Software Technology, Nanjing University
[2] HKUST HKUST(GZ)  [3] Tsinghua University  [4] The University of Queensland

## Abstract

Graph anomaly detection (GAD) has garnered increasing attention in recent years, yet remains challenging due to two key factors: (1) label scarcity stemming from the high cost of annotations and (2) homophily disparity at node and class levels. In this paper, we introduce Anomaly-Aware Pre-Training and Fine-Tuning (APF), a targeted and effective framework to mitigate the above challenges in GAD. In the pre-training stage, APF incorporates node-specific subgraphs selected via the Rayleigh Quotient, a label-free anomaly metric, into the learning objective to enhance anomaly awareness. It further introduces two learnable spectral polynomial filters to jointly learn dual representations that capture both general semantics and subtle anomaly cues. During fine-tuning, a gated fusion mechanism adaptively integrates pre-trained representations across nodes and dimensions, while an anomaly-aware regularization loss encourages abnormal nodes to preserve more anomaly-relevant information. Furthermore, we theoretically show that APF tends to achieve linear separability under mild conditions. Comprehensive experiments on 10 benchmark datasets validate the superior performance of APF in comparison to state-of-the-art baselines. The code is available at https://github.com/Cloudy1225/APF.

## 1 Introduction

Graph anomaly detection (GAD) aims to identify a small but significant portion of instances, such as abnormal nodes, that deviate significantly from the standard, normal, or prevalent patterns within graph-structured data (Qiao et al., 2025b). The detection of these anomalies is crucial for various scenarios, such as financial fraud in transaction networks (Cheng et al., 2025), fake news in social media (Aïmeur et al., 2023), and sensor faults in IoT networks (Gaddam et al., 2020). Given their strong ability to model relational structure, graph neural networks (GNNs) have recently emerged as a leading choice for tackling GAD.

Despite notable progress, most existing GAD methods (Dou et al., 2020; Tang et al., 2022; Zhuo et al., 2024; Zheng et al., 2025a) are not tailored for label-scarce scenarios, leading to suboptimal performance in real-world deployments where annotations are costly. Recent semi-supervised attempts, such as pseudo-labeling (Rizve et al., 2021; Chen et al., 2024b) and synthetic sample generation (Ma et al., 2024; Qiao et al., 2024), seek to mitigate this but often suffer from instability due to the inherent uncertainty and confirmation bias that compromise performance (Rizve et al., 2021). In contrast, the pre-training and fine-tuning paradigm has shown great promise in label-scarce learning across CV (Chen et al., 2020; Nandam et al., 2025), NLP (Devlin et al., 2019; Shi et al., 2023), and graph learning (Zhu et al., 2020; Ju et al., 2024). A recent study (Cheng et al., 2024) further reveals that general-purpose graph pre-training strategies (Veličković et al., 2019; Hou et al., 2022) can match or even outperform GAD-specific methods under limited supervision. Despite their strong potential, existing graph pre-training frameworks are primarily designed to extract task-agnostic semantic knowledge. As such, they fail to address GAD's unique challenges, falling short in capturing anomaly-relevant cues and leaving their adaptation to GAD an unsolved yet pressing task.

---

*Equal contribution.
†Corresponding author (hetieke@gmail.com).

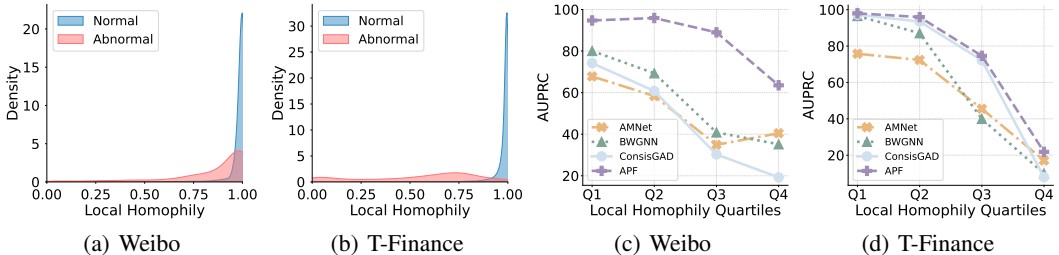

Figure 1: (a), (b): Distribution of local homophily for Weibo and T-Finance. (c), (d): Performance across local homophily quartiles (Q1 = top 25%, Q4 = bottom 25%) on Weibo and T-Finance.

In GAD, global homophily, the intra-class edge ratio over the entire graph, tends to decrease due to challenges like camouflage (Dou et al., 2020), making it an intuitive yet common anomaly indicator. Nonetheless, we highlight that local homophily, the class consistency within each node's neighborhood, reveals more nuanced disparity patterns, capturing localized yet often overlooked anomaly signals. As illustrated in Figures 1(a) and 1(b), these disparities manifest at two granularities: **(1) node-level disparity** represents the high variations in local homophily across individual nodes and **(2) class-level disparity** refers to lower local homophily for abnormal nodes. Most existing approaches are built around global homophily and employ uniform processing schemes, such as edge reweighting (Shi et al., 2022; Gao et al., 2023) or spectral filtering (Tang et al., 2022; Zheng et al., 2025a). However, such globally uniform designs lack node-adaptive mechanisms necessary to accommodate the structural diversity of individual nodes, resulting in inconsistent anomaly distinguishability across nodes in different local homophily groups, as further evidenced in Figures 1(c) and 1(d). These limitations jointly highlight a critical challenge: *How can we devise a GAD-specific pre-training and fine-tuning framework that effectively mitigates dual-granularity local homophily disparity?*

**Present Work.** To address this challenge, we introduce Anomaly-Aware Pre-Training and Fine-Tuning (APF), a novel framework designed for GAD under limited supervision, grounded in anomaly-aware pre-training and granularity-adaptive fine-tuning to handle homophily disparity.

In the context of anomaly-aware pre-training, APF harnesses the Rayleigh Quotient, a label-free metric for quantifying anomaly degree, to reduce reliance on label-dependent anomaly measures. In particular, building upon conventional pre-training objectives, we incorporate node-wise subgraphs, each selected to maximize the Rayleigh Quotient, into the objective to enhance anomaly awareness. We further adopt learnable spectral polynomial filters to jointly optimize two distinct representations: one capturing general semantic patterns and the other focusing on subtle anomaly cues. This dual-objective design effectively captures node-wise structural disparity, offering more informative initializations for downstream detection tasks.

To enable granularity-adaptive fine-tuning, APF employs a gated fusion network that adaptively combines pre-trained representations at both the node and dimension levels. An anomaly-aware regularization loss is further introduced to encourage abnormal nodes to retain more anomaly-relevant information from pre-trained representations than normal nodes. Together, these components explicitly address local homophily disparity, by aligning the fine-tuning with the homophily disparity at node and class levels under label-guided optimization. Theoretical analysis further shows that APF tends to achieve linear separability across all nodes.

Empirically, extensive experiments on 10 benchmark datasets are conducted to verify the superior performance of APF, demonstrating its effectiveness against label scarcity.

## 2 PRELIMINARY

In this section, we briefly introduce notations and key concepts. Detailed preliminaries and related works are provided in Appendix B and Appendix C, respectively.

**Graph Anomaly Detection (GAD).** Let $\mathcal{G} = (\mathcal{V}, \mathcal{E}, \boldsymbol{X})$ be a graph with $n$ nodes and edges $\mathcal{E}$. Each node $v_i$ has a $d$-dimensional feature $\boldsymbol{x}_i$, forming the feature matrix $\boldsymbol{X} \in \mathbb{R}^{n \times d}$. The adjacency matrix

is $\boldsymbol{A}$, and $\boldsymbol{D}$ is the diagonal degree matrix. The neighbor set of $v_i$ is denoted as $\mathcal{N}_i$. GAD is framed as a binary classification problem where anomalies are regarded as positive with label 1. Given labeled nodes $\mathcal{V}^L = \mathcal{V}_a \cup \mathcal{V}_a$ with labels $\boldsymbol{y}^L$, the goal is to predict $\hat{\boldsymbol{y}}^U$ for unlabeled nodes. Real-world settings typically exhibit extreme label scarcity ($|\mathcal{V}^L| \ll |\mathcal{V}|$) and class imbalance ($|\mathcal{V}_a| \ll |\mathcal{V}_n|$).

**Local Homophily.** Given a node $v_i$, its local homophily $h_i$ is defined as the fraction of neighbors in $\mathcal{N}_i$ that share the same label:

$$h_i = \frac{|v_j \in \mathcal{N}_i : y_i = y_j|}{|\mathcal{N}_i|}. \tag{1}$$

We then compute the average local homophily of abnormal and normal nodes, denoted by $h^a$ and $h^n$, respectively, as:

$$h^a = \frac{\sum_{y_i=1} h_i}{\sum_{y_i=1} 1}, \quad h^n = \frac{\sum_{y_i=0} h_i}{\sum_{y_i=0} 1}. \tag{2}$$

Both metrics are bounded within $[0, 1]$. Table 4 summarizes the values of $h^a$ and $h^n$ across datasets, where $h^a$ is consistently lower than $h^n$, highlighting the presence of class-level homophily disparity.

**Graph Spectral Filtering.** Given the adjacency matrix $\boldsymbol{A}$, the graph Laplacian is defined as $\boldsymbol{L} = \boldsymbol{D} - \boldsymbol{A}$, which is symmetric and positive semi-definite. Their symmetric normalized versions are noted as $\tilde{\boldsymbol{A}} = \boldsymbol{D}^{-1/2}\boldsymbol{A}\boldsymbol{D}^{-1/2}$ and $\tilde{\boldsymbol{L}} = \boldsymbol{I} - \boldsymbol{D}^{-1/2}\boldsymbol{A}\boldsymbol{D}^{-1/2}$. It admits eigendecomposition $\tilde{\boldsymbol{L}} = \boldsymbol{U}\boldsymbol{\Lambda}\boldsymbol{U}^\top$, where $\boldsymbol{U} \in \mathbb{R}^{n \times n}$ contains orthonormal eigenvectors (the graph Fourier basis) and $\boldsymbol{\Lambda} = \mathrm{diag}(\lambda_1, \cdots, \lambda_n)$ are eigenvalues ordered as $0 = \lambda_1 \le \cdots \le \lambda_n \le 2$ (frequencies). The graph spectral filtering is then defined as $\hat{\boldsymbol{X}} = \boldsymbol{U}g(\boldsymbol{\Lambda})\boldsymbol{U}^\top\boldsymbol{X}$, where $g(\cdot)$ denotes the spectral filter.

## 3 METHODOLOGY

In this section, we formally introduce our APF framework, a targeted and effective solution tailored to the unique challenges in GAD. The overall framework is illustrated in Figure 2.

### 3.1 ANOMALY-AWARE PRE-TRAINING

**Label-free Anomaly Indicator.** Most existing graph pre-training strategies (Veličković et al., 2019; Hou et al., 2022; Liu et al., 2024d) are designed to optimize task-agnostic objectives. As a result, the learned representations primarily encode general semantic knowledge. In contrast, the goal of GAD is to capture subtle anomaly cues that differentiate minority abnormal instances. However, many of these cues, such as relation camouflage (Dou et al., 2020), are inherently label-dependent, making them difficult to capture under the label-free pre-training.

This motivates the incorporation of effective label-free anomaly measures into the learning objective to guide the representation toward anomaly-aware semantics. Recent findings reveal that the existence of anomalies induces the 'right-shift' phenomenon (Tang et al., 2022; Gao et al., 2023; Dong et al., 2024), where spectral energy distribution concentrates more on high frequencies. The corresponding accumulated spectral energy can be quantified by the Rayleigh Quotient (Horn & Johnson, 2012):

$$RQ(\boldsymbol{x}, \boldsymbol{L}) = \frac{\boldsymbol{x}^T \boldsymbol{L} \boldsymbol{x}}{\boldsymbol{x}^T \boldsymbol{x}} = \frac{\sum_{i=1}^n \lambda_i \hat{x}_i^2}{\sum_{i=1}^n \hat{x}_i^2} = \frac{\sum_{i,j} A_{ij}(x_j - x_i)^2}{\sum_{i=1}^n x_i^2}, \tag{3}$$

where $\boldsymbol{x} \in \mathbb{R}^n$ represents a general graph signal and $\hat{\boldsymbol{x}} = \boldsymbol{U}^T\boldsymbol{x}$ denotes its projection in spectral space. Crucially, the Rayleigh Quotient measures the inconsistency between node attributes and the local graph structure (i.e., graph smoothness). A high value indicates that connected nodes have dissimilar features, which allows RQ to capture both *attribute anomalies* (where features deviate from neighbors) and *structural anomalies* (e.g., camouflaged edges connecting dissimilar nodes) (Tang et al., 2022). We provide a detailed discussion on how RQ captures these different anomaly types in Appendix K. The following lemma further illustrates the relationship between the Rayleigh Quotient and anomaly information.

**Lemma 1.** *The Rayleigh Quotient $RQ(x, L)$, which represents the accumulated spectral energy of a graph signal, increases monotonically with the anomaly degree. (Tang et al., 2022)*

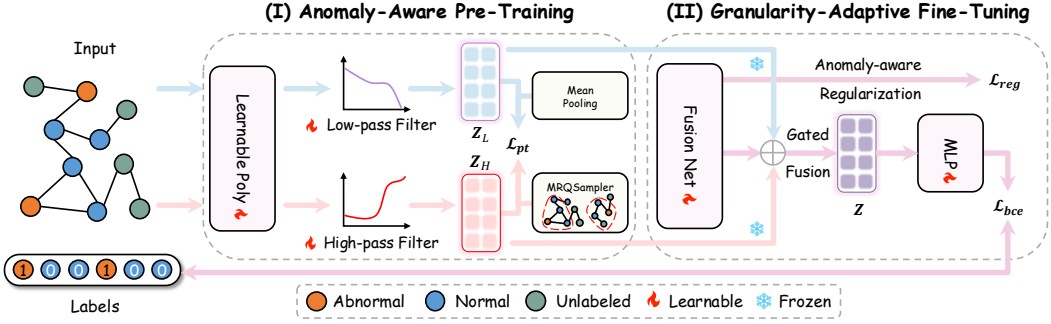

Figure 2: Overview of our proposed APF.

Given this, Rayleigh Quotient stands out as a promising label-free anomaly metric. To guide each node $v_i$ in capturing its potential anomaly cues, we employ MRQSampler (Lin et al., 2024) to extract its 2-hop subgraph $\mathcal{G}_i^{RQ}$. Each subgraph is selected to maximize the Rayleigh Quotient, thereby preserving as much structural diversity and anomaly-relevant signal as possible (Lin et al., 2024).

**Dual-filter Encoding.** Beyond the learning objective, the architectural modules for encoding anomaly-relevant information are equally critical. Recent studies highlight that capturing high-frequency components is essential for modeling heterophilic patterns (Bo et al., 2021; Luan et al., 2022; Lei et al., 2022). Motivated by this, we complement the conventional low-pass encoder with an additional high-pass encoder. This dual-branch design allows us to simultaneously capture general semantic patterns (via low-pass filters) and subtle anomaly cues (via high-pass filters), the latter of which are often smoothed out by standard GNNs. We provide a detailed discussion on the motivation and spectral guarantees of this design in Appendix L. To facilitate flexible spectral encoding, we adopt the learnable $K$-order Chebyshev polynomial (He et al., 2022) and restrict it to fit only low-pass and high-pass filters, denoted by $g_L(\cdot)$ and $g_H(\cdot)$, following prior work Chen et al. (2024a):

$$g_L(\hat{\boldsymbol{L}}) = \sum_{k=0}^{K} w_k^L T_k(\hat{\boldsymbol{L}}), \; g_H(\hat{\boldsymbol{L}}) = \sum_{k=0}^{K} w_k^H T_k(\hat{\boldsymbol{L}}), \tag{4}$$

where $\hat{\boldsymbol{L}} = 2\tilde{\boldsymbol{L}}/\lambda_n - \boldsymbol{I}$ denotes the scaled Laplacian matrix. The Chebyshev polynomials are recursively defined as $T_k(x) = 2xT_{k-1}(x) - T_{k-2}(x)$ with $T_0(x) = 1$ and $T_1(x) = x$. The coefficients are computed as:

$$w_k^L = \frac{2}{M+1} \sum_{i=1}^{K} \gamma_i^L T_k(t_i), \; w_k^H = \frac{2}{M+1} \sum_{i=1}^{K} \gamma_i^H T_k(t_i), \tag{5}$$

where $t_i = \cos(\frac{i+1/2}{K+1}\pi), i = 0, \cdots, K$ denotes the Chebyshev nodes for $T_{K+1}(x)$. The filter values $\gamma_m^L$ and $\gamma_m^H$ is determined by:

$$\gamma_k^L = \gamma_0 - \sum_{j=1}^{k} \gamma_j, \; \gamma_k^H = \sum_{j=0}^{k} \gamma_j, \tag{6}$$

where $\boldsymbol{\gamma} = (\gamma_0, \cdots, \gamma_M)$ is the shared learnable parameter with $\gamma_0 = \gamma_0^L = \gamma_0^H$. As per Chen et al. (2024a), we have $\gamma_i^H \leq \gamma_{i+1}^H$ and $\gamma_i^L \geq \gamma_{i+1}^L$, thus guarantee the high-pass/low-pass property for $g_L(\cdot)/g_H(\cdot)$. Given such, our low-pass and high-pass encoders are formulated as:

$$\boldsymbol{Z}_L = f_{\theta_L}\left(g_L(\hat{\boldsymbol{L}})\boldsymbol{X}\right), \; \boldsymbol{Z}_H = f_{\theta_H}\left(g_H(\hat{\boldsymbol{L}})\boldsymbol{X}\right), \tag{7}$$

where $\boldsymbol{Z}_L, \boldsymbol{Z}_H \in \mathbb{R}^{n \times e}$ denote the low-pass and high-pass node representations respectively. $f_{\theta_L}(\cdot), f_{\theta_H}(\cdot)$ represent the learnable MLP for each filter. To ensure consistent scaling for subsequent fusion, both $\boldsymbol{Z}_L$ and $\boldsymbol{Z}_H$ are standardized with zero mean and unit variance.

**Optimization.** Unlike conventional objectives that only preserve task-agnostic semantics, our optimization explicitly integrates anomaly awareness. We build the pre-training framework upon DGI (Veličković et al., 2019) owing to its efficiency and effectiveness: we retain its standard low-pass branch to encode generic semantic structure, and further incorporate an anomaly-sensitive objective that maximizes the mutual information between each node and the summary of its Rayleigh Quotient-guided subgraph, computed by high-pass encoders. Let $\tilde{\boldsymbol{Z}}_L$ and $\tilde{\boldsymbol{Z}}_H$ denote negative samples generated from randomly shuffled inputs (Veličković et al., 2019). Our pre-training loss is:

$$\mathcal{L}_{pt} = -\frac{1}{n}\sum_i^n \left(\log \mathcal{D}\left(\boldsymbol{Z}_i^L, \boldsymbol{s}^L\right) + \log\left(1 - \mathcal{D}\left(\tilde{\boldsymbol{Z}}_i^L, \boldsymbol{s}^L\right)\right)\right)$$
$$-\frac{1}{n}\sum_i^n \left(\log \mathcal{D}\left(\boldsymbol{Z}_i^H, \boldsymbol{s}_i^H\right) + \log\left(1 - \mathcal{D}\left(\tilde{\boldsymbol{Z}}_i^H, \boldsymbol{s}_i^H\right)\right)\right), \quad (8)$$

where $\boldsymbol{s}^L = \frac{1}{n}\sum_{i=1}^n \boldsymbol{Z}_i^L$ is the global semantic summary and $\boldsymbol{s}_i^H = \frac{1}{|\mathcal{G}_i^{RQ}|}\sum_{v_j \in \mathcal{G}_i^{RQ}} \boldsymbol{Z}_j^H$ is the anomaly-aware summary over the Rayleigh Quotient-based subgraph of node $v_i$. The discriminator is defined as $\mathcal{D}(\boldsymbol{z}, \boldsymbol{s}) = \sigma(\boldsymbol{z}^\top \boldsymbol{W}\boldsymbol{s})$. By jointly optimizing these two complementary objectives, our framework departs from existing pre-training methods and introduces a principled way to capture task-agnostic semantic knowledge and node-specific anomaly cues, yielding more informative representations for downstream anomaly detection.

## 3.2 GRANULARITY-ADAPTIVE FINE-TUNING

**Node- and Dimension-wise Fusion.** After pre-training, we aim to develop a node-adaptive fusion mechanism that selectively combines task-agnostic semantic knowledge ($\boldsymbol{Z}_L$) and node-specific structural disparities ($\boldsymbol{Z}_H$) from the frozen pre-trained representations. This fusion is designed to better respond to local homophily variations across nodes under label-guided learning. Beyond trivial node-wise fusion, prior studies (Wang & Zhang, 2022; Dong et al., 2021; Zheng et al., 2025b) highlight that different feature dimensions contribute unequally to downstream tasks, motivating a dimension-aware fusion design. To this end, we introduce a coefficient matrix $\boldsymbol{C} \in [0,1]^{n \times e}$ to combine $\boldsymbol{Z}_L$ and $\boldsymbol{Z}_H$ at node and dimension levels:

$$\boldsymbol{Z} = \boldsymbol{C} \odot \boldsymbol{Z}_L + (1 - \boldsymbol{C}) \odot \boldsymbol{Z}_H, \quad (9)$$

where $\odot$ denotes the Hadamard product, and $\boldsymbol{Z}$ is the resulting fused representation passed to the classifier. A naive approach to learn $\boldsymbol{C}$ is treating it as free parameters (Wang & Zhang, 2022), but this leads to excessive overhead ($\mathcal{O}(n \times e)$), inefficient learning under sparse supervision, and neglect of input semantics. To alleviate this, we introduce a lightweight Gated Fusion Network (GFN) that generates coefficients based on the input features:

$$\boldsymbol{C} = \sigma\left(\boldsymbol{X}\boldsymbol{W}_c + \boldsymbol{b}_c\right), \quad (10)$$

where $\boldsymbol{W}_c \in \mathbb{R}^{d \times e}$ is a learnable matrix and $\boldsymbol{b}_c \in \mathbb{R}^e$ is a bias term, $\sigma(\cdot)$ is sigmoid function. The advantages of GFN over direct optimization are multifold: **(1)** GFN reduces the learnable parameter complexity of $\boldsymbol{C}$ from $\mathcal{O}(n \times e)$ to $\mathcal{O}((d+1) \times e)$, where $d, e \ll n$; **(2)** GFN allows sparse supervision to update the entire coefficient matrix, while direct optimization only affects labeled rows; **(3)** GFN leverages raw input features, which encode valuable anomaly-relevant attributes (Tang et al., 2023).

**Anomaly-aware Regularization Loss.** As indicated by the class-level local homophily disparity, abnormal nodes tend to camouflage themselves by connecting to normal ones. To account for this behavior, they should rely less on generic knowledge and be assigned more anomaly-relevant cues from pre-trained representations during the fusion process. To encourage this class-specific fusion preference, we introduce a regularization term that guides the optimization of the coefficient matrix $\boldsymbol{C}$ accordingly. Let $c_i = \frac{1}{e}\sum_{j=1}^e \boldsymbol{C}_{ij}$ denote the average fusion weight of node $v_i$ toward generic knowledge. We encourage $c_i$ to approach a class-specific target: $p^a \in [0,1]$ for abnormal nodes and $p^n \in [0,1]$ for normal nodes, with the constraint $p^a \leq p^n$, to mimic the observed class-level disparity.

The regularization loss is formulated as a binary cross-entropy loss:

$$\mathcal{L}_{reg} = -\frac{1}{|\mathcal{V}^L|} \sum_{v_i \in \mathcal{V}^L, y_i=1} (p^a \log c_i + (1 - p^a) \log(1 - c_i))$$
$$- \frac{1}{|\mathcal{V}^L|} \sum_{v_i \in \mathcal{V}^L, y_i=0} (p^n \log c_i + (1 - p^n) \log(1 - c_i)). \tag{11}$$

This encourages the model to incorporate class-level fusion bias and enhances its ability to distinguish anomalies.

**Optimization.** Given the fused representations $\boldsymbol{Z}$ from Eq. 9, we employ a two-layer MLP to predict the label $\hat{y}_i$ for each labeled node $v_i$. The model is optimized using the standard binary cross-entropy loss:

$$\mathcal{L}_{bce} = -\frac{1}{|\mathcal{V}^L|} \sum_{i \in \mathcal{V}^L} (y_i \log \hat{y}_i + (1 - y_i) \log(1 - \hat{y}_i)). \tag{12}$$

The overall fine-tuning objective combines the classification loss with the regularization term:

$$\mathcal{L}_{ft} = \mathcal{L}_{bce} + \mathcal{L}_{reg}. \tag{13}$$

## 3.3 THEORETICAL INSIGHTS

In this subsection, we provide a theoretical analysis to support our architectural design. Our theoretical analysis is grounded in the Contextual Stochastic Block Model (Deshpande et al., 2018), a widely used generative model for attributed graphs (Baranwal et al., 2021; Ma et al., 2022; Mao et al., 2023; Han et al., 2024). To properly reflect homophily disparity, degree heterogeneity, and class imbalance in GAD, we first introduce a variant, the Anomalous Stochastic Block Model (ASBM).

**Definition 1** ($ASBM(n_a, n_n, \boldsymbol{\mu}, \boldsymbol{\nu}, (p_1, q_1), (p_2, q_2), \boldsymbol{\theta})$). *Let $\mathcal{C}_a$ and $\mathcal{C}_n$ be the abnormal and normal node sets with sizes $n_a$ and $n_n$, respectively; $n = n_a + n_n$, class priors $\pi_a = n_a/n$, $\pi_n = 1 - \pi_a$ with $\pi_a \ll \pi_n$. Node features are sampled row-wise as $\boldsymbol{X}_a \sim \mathcal{N}(\boldsymbol{\mu}, \frac{1}{d}\boldsymbol{I})$ and $\boldsymbol{X}_n \sim \mathcal{N}(\boldsymbol{\nu}, \frac{1}{d}\boldsymbol{I})$ with $\|\boldsymbol{\mu}\|_2, \|\boldsymbol{\nu}\|_2 \leq 1$, and the full matrix $\boldsymbol{X} = [\boldsymbol{X}_a; \boldsymbol{X}_n]$. Each node $v_i$ has a random variable $\theta_i > 0$ (collected in $\boldsymbol{\theta}$) as the degree parameter, which controls how many edges it tends to form. Nodes can follow either a homophilic or heterophilic connectivity pattern: a node in the homophilic set $\mathcal{H}_o$ prefers to connect to nodes of the same class, while a node in the heterophilic set $\mathcal{H}_e$ prefers the opposite. Accordingly, edges are generated independently as follows:*

$$\mathbb{P}(A_{ij} = 1 \mid y_i, y_j, h_i) = \theta_i \theta_j \times \begin{cases} p_1 & \text{if } h_i = ho \text{ and } y_i = y_j, \quad p_2 & \text{if } h_i = ho \text{ and } y_i = y_j, \\ q_1 & \text{if } h_i = he \text{ and } y_i \neq y_j, \quad q_2 & \text{if } h_i = he \text{ and } y_i \neq y_j. \end{cases}$$

*Here $y_i \in \{a, n\}$ is the class label and $h_i \in \{ho, he\}$ indicates the local pattern (homophilic/heterophilic) adopted by node $v_i$; $p_1 > q_1$ enforces homophily (intra-class edges more likely), while $p_2 < q_2$ enforces heterophily (inter-class edges more likely).*

In line with prior analyses (Baranwal et al., 2021), we adopt a linear classifier parameterized by $\boldsymbol{w} \in \mathbb{R}^d$ and $b \in \mathbb{R}$, with predictions $\hat{\boldsymbol{y}} = \sigma(\hat{\boldsymbol{X}}\boldsymbol{w} + b\boldsymbol{1})$ on frozen filtered features $\hat{\boldsymbol{X}} = \boldsymbol{U}g(\boldsymbol{\Lambda})\boldsymbol{U}^\top \boldsymbol{X}$, optimized by the binary cross-entropy in Eq. 12. The separability of this model under node-adaptive filtering is characterized by the following theorem:

**Theorem 1.** *For a graph $\mathcal{G}(\mathcal{V}, \mathcal{E}, \boldsymbol{X}) \sim ASBM(n_a, n_n, \boldsymbol{\mu}, \boldsymbol{\nu}, (p_1, q_1), (p_2, q_2), \boldsymbol{\theta})$, when low- and high-pass filters are applied separately to the homophilic and heterophilic node sets $\mathcal{H}_o, \mathcal{H}_e$, there exist parameters $\boldsymbol{w}^*, b^*$ such that all nodes are linearly separable with probability $1 - o_d(1)$.*

We present the detailed proof in Appendix E. This theorem theoretically establishes that, under appropriate conditions, adaptively applying low-pass and high-pass filters to nodes based on local homophily is possible to achieve linear separability across all nodes.

While the theorem assumes an idealized scenario with oracle filter assignments, our architectural design implements this principle in a data-driven manner. APF first extracts candidate representations via both low-pass and high-pass filters. Then, instead of hard filter selection, it employs a Gated

Fusion Network to generate soft, continuous coefficients that approximate the node-wise filter assignment. Guided by class-specific fusion preferences and classification loss, our model learns to sense local homophily disparity and adjust fusion weights accordingly. This enables the learned fusion strategy to mimic the theoretical node-wise filter assignment without requiring all nodes' pattern labels, allocating representations based on each node's local homophily, thereby approaching the theoretical bound of linear separability and ultimately improving GAD performance.

# 4 EXPERIMENTS

In this section, we first evaluate the effectiveness of pre-training for GAD, the overall performance of our model, and its ability to mitigate homophily disparity (Section 4.2). We then analyze the contribution of each component and provide visualizations of the learned fusion coefficients (Section 4.3). Due to space constraints, additional experimental results, such as efficiency comparison, hyperparameter analysis, and representation visualization, are presented in Appendix I.

## 4.1 EXPERIMENTAL SETUP

**Datasets and Baselines.** We conduct experiments on 10 GADBench (Tang et al., 2023) datasets spanning diverse domains and scales. We compare with a broad range of baseline methods, including standard GNNs (Kipf & Welling, 2017; Xu et al., 2019; Veličković et al., 2018; Luan et al., 2022; Bo et al., 2021; Dong et al., 2021; He et al., 2021), GAD-specific models (Li et al., 2019; Wang et al., 2021; Liu et al., 2021a; Chai et al., 2022; Tang et al., 2022; Gao et al., 2023; Chen et al., 2024b; Dong et al., 2025), and graph pre-training approaches (Veličković et al., 2019; Zhu et al., 2020; Bielak et al., 2022; Hou et al., 2022; Thakoor et al., 2022; Liu et al., 2024c; Chen et al., 2024a). For detailed dataset statistics, dataset descriptions, and baseline descriptions, please kindly see Table 4, Appendix H.1, and Appendix H.2, respectively.

**Metrics and Implementation Details.** To strictly align with real-world label scarcity, we follow the **semi-supervised setting** defined in GADBench (Tang et al., 2023). Specifically, we standardize the training set to include exactly 100 labeled nodes (20 anomalies and 80 normal nodes) across all datasets. Thus, the baseline results reported in this paper correspond to the semi-supervised performance metrics found in the GADBench Appendix (Table 11). For evaluation, we employ AUPRC, AUROC, and Rec@K. We prioritize AUPRC over the threshold-dependent F1 score to ensure a robust assessment of precision-recall tradeoffs independent of decision thresholds, which can be unstable under limited supervision. All experiments are averaged over 10 random splits provided by GADBench for robustness. Due to space constraints, detailed evaluation protocols and hyperparameter settings are provided in Appendix H.3 and Appendix H.4, respectively. Our code is available at `https://github.com/Cloudy1225/APF`.

## 4.2 PERFORMANCE COMPARISON

We summarize the performance across AUPRC, AUROC, and Rec@K in Table 1, Table 7, and Table 8, respectively. For comprehensiveness, DGI and BWDGI represent standard DGI pre-training with GCN and BWGNN as backbones. In addition, APF (*w/o* $\mathcal{L}_{pt}$) is a variant of our method, skipping pre-training and directly optimizing the learnable spectral filters using $\mathcal{L}_{ft}$.

**Effectiveness of Pre-Training.** Overall, pre-training methods (from DGI to BWDGI and APF) achieve competitive or even superior performance compared to GAD-specific models, especially on Reddit, Weibo, and Tolokers. In particular, DGI, BWDGI, and APF bring AUPRC improvements of +7.3%, +3.8%, and +4.9% over their respective end-to-end training counterparts (GCN, BWGNN, and APF (*w/o* $\mathcal{L}_{pt}$)). These gains highlight the importance of pre-training in addressing the label scarcity inherent to GAD, by providing more expressive and transferable initializations.

To further validate the benefits of our pre-training, we also evaluate the learned representations in a purely unsupervised setting, where they are directly coupled with an anomaly scorer IF (Liu et al., 2008) without fine-tuning. The results, presented in Appendix I.1, confirm that our pre-training further enhances unsupervised detection performance, demonstrating its general effectiveness.

Table 1: Comparison of AUPRC for each model. "-" denotes "out of memory". The best and runner-up models are **bolded** and underlined.

| Model | Reddit | Weibo | Amazon | Yelp | T-Fin. | Ellip. | Tolo. | Quest. | DGraph. | T-Social | Avg. |
|---|---|---|---|---|---|---|---|---|---|---|---|
| GCN | 4.2±0.8 | 86.0±6.7 | 32.8±1.2 | 16.4±2.6 | 60.5±10.8 | 43.1±4.6 | 33.0±3.6 | 6.1±0.9 | 2.3±0.2 | 8.4±3.8 | 29.3 |
| GIN | 4.3±0.6 | 67.6±7.4 | 75.4±4.3 | 23.7±5.4 | 44.8±7.1 | 40.1±3.2 | 31.8±3.2 | 6.7±1.1 | 2.0±0.1 | 6.2±1.7 | 30.3 |
| GAT | 4.7±0.7 | 73.3±7.3 | 81.6±1.7 | 25.0±2.9 | 28.9±8.6 | 44.2±6.6 | 33.0±2.0 | 7.3±1.2 | 2.2±0.2 | 9.2±2.0 | 30.9 |
| ACM | 4.4±0.7 | 66.0±8.7 | 54.0±19.0 | 21.4±2.7 | 29.2±16.8 | 63.1±4.8 | 34.4±3.5 | 7.2±1.9 | 2.2±0.4 | 6.0±1.6 | 28.8 |
| FAGCN | 4.7±0.7 | 70.1±10.6 | 77.0±2.3 | 22.5±2.6 | 39.8±27.2 | 43.6±10.6 | 35.0±4.3 | 7.3±1.4 | 2.0±0.3 | - | - |
| AdaGNN | 4.9±0.8 | 28.3±2.8 | 75.7±6.3 | 22.7±2.1 | 23.3±7.6 | 39.2±7.9 | 32.2±3.9 | 5.3±0.9 | 2.1±0.3 | 4.8±1.1 | 23.9 |
| BernNet | 4.9±0.3 | 66.6±5.5 | 81.2±2.4 | 23.9±2.7 | 51.8±12.4 | 40.0±4.1 | 28.9±3.5 | 6.7±2.1 | 2.5±0.2 | 4.2±1.2 | 31.1 |
| GAS | 4.7±0.7 | 65.7±8.4 | 80.7±1.7 | 21.7±3.3 | 45.7±13.4 | 46.0±4.9 | 31.7±3.0 | 6.3±2.0 | 2.5±0.2 | 8.6±2.4 | 31.4 |
| DCI | 4.3±0.4 | 76.2±4.3 | 72.5±7.9 | 24.0±4.8 | 51.0±7.2 | 43.4±4.9 | 32.1±4.2 | 6.1±1.3 | 2.0±0.2 | 7.4±2.5 | 31.9 |
| PCGNN | 3.4±0.5 | 69.3±9.7 | 81.9±1.9 | 25.0±3.5 | 58.1±11.3 | 40.3±6.6 | 33.9±1.7 | 6.4±1.8 | 2.4±0.4 | 8.0±1.6 | 32.9 |
| AMNet | 4.9±0.4 | 67.1±5.1 | 82.4±2.2 | 23.9±3.5 | 60.2±8.2 | 33.3±4.8 | 28.6±1.5 | 7.4±1.4 | 2.2±0.3 | 3.1±0.3 | 31.3 |
| BWGNN | 4.2±0.7 | 80.6±4.7 | 81.7±2.2 | 23.7±2.9 | 60.9±13.8 | 43.4±5.5 | 35.3±2.2 | 6.5±1.7 | 2.1±0.3 | 15.9±6.2 | 35.4 |
| GHRN | 4.2±0.6 | 77.0±6.2 | 80.7±1.7 | 23.8±2.8 | 63.4±10.4 | 44.2±5.7 | 35.9±2.0 | 6.5±1.7 | 2.3±0.2 | 16.2±4.6 | 35.4 |
| ConsisGAD | 4.5±0.5 | 64.6±5.5 | 78.7±5.7 | 25.9±2.9 | 79.7±4.7 | 47.8±8.2 | 33.7±2.7 | 7.9±2.4 | 2.0±0.2 | 41.3±5.0 | 38.6 |
| SpaceGNN | 4.6±0.5 | 79.2±2.8 | 81.1±2.3 | 25.7±2.4 | 81.0±3.5 | 44.1±3.5 | 33.8±2.5 | 7.4±1.6 | 2.0±0.3 | 59.0±5.7 | 41.8 |
| XGBGraph | 4.1±0.5 | 75.9±6.2 | **84.4±1.1** | 24.8±3.1 | 78.3±3.1 | **77.2±3.2** | 34.1±2.8 | 7.7±2.1 | 1.9±0.2 | 40.6±7.6 | 42.9 |
| DGI | 4.8±0.6 | 90.8±2.5 | 46.5±3.7 | 17.0±1.2 | 75.0±4.9 | 45.9±2.5 | 39.7±0.8 | 6.4±1.2 | 2.1±0.2 | 37.8±6.1 | 36.6 |
| GRACE | 4.7±0.3 | 90.8±1.8 | 51.3±4.3 | 18.2±1.6 | 79.3±0.7 | 48.1±3.6 | 37.4±2.8 | 8.9±1.7 | - | - | - |
| G-BT | 5.0±0.7 | 87.5±3.9 | 38.7±2.2 | 18.8±1.6 | 76.8±1.8 | 45.2±4.4 | 37.9±2.8 | 9.1±1.9 | 2.6±0.3 | 42.2±7.4 | 36.4 |
| GraphMAE | 4.3±0.1 | 91.4±2.6 | 39.4±0.3 | 17.3±0.1 | 70.8±4.7 | 32.7±3.8 | 36.0±2.1 | 5.9±0.5 | 2.1±0.1 | 42.6±10.5 | 34.2 |
| BGRL | 5.3±0.3 | 93.6±1.9 | 43.9±4.6 | 19.2±1.6 | 61.7±5.1 | 47.4±6.0 | 38.3±3.3 | 8.4±2.0 | 2.0±0.2 | 46.7±8.4 | 36.6 |
| SSGE | 4.8±0.9 | 87.7±2.6 | 39.1±2.4 | 18.8±1.6 | 77.6±0.9 | 47.4±3.0 | 38.2±2.8 | 8.1±1.1 | 2.5±0.3 | 46.4±3.9 | 37.1 |
| PolyGCL | 5.2±0.8 | 87.3±2.1 | 79.7±6.6 | 24.3±2.5 | 43.3±6.4 | 50.0±5.2 | 33.0±1.8 | 5.7±0.8 | 2.2±0.3 | 40.6±7.0 | 37.1 |
| BWDGI | 4.5±0.6 | 72.5±2.7 | 79.4±5.7 | 26.8±2.7 | 80.0±2.1 | 44.9±6.5 | 38.5±3.1 | 5.0±0.7 | 2.4±0.2 | 38.0±5.2 | 39.2 |
| APF (*w/o* $\mathcal{L}_{pt}$) | 5.2±0.6 | 85.8±7.9 | 82.7±3.0 | 24.1±2.2 | 79.4±3.4 | 55.5±4.9 | 37.4±1.2 | 9.4±1.5 | 2.3±0.2 | 64.8±10.5 | 44.7 |
| **APF** | **5.9±0.9** | **93.9±1.1** | 83.8±2.9 | **28.4±1.4** | **82.5±2.6** | 67.7±3.4 | **40.5±2.0** | **12.3±1.6** | **2.9±0.2** | **77.8±5.6** | **49.6** |

**Superiority of Our Proposed APF.** As observed, APF outperforms both GAD-specific approaches and graph pre-training methods in most cases. On average, APF achieves gains of +6.7% in AUPRC, +3.8% in AUROC, and +6.2% in Rec@K. Even without the pre-training, APF (*w/o* $\mathcal{L}_{pt}$) surpasses all baseline methods, including BWGNN and AMNet, which also adopt multi-filter architectures. This highlights the strength of our fine-tuning module in adaptively emphasizing anomaly-relevant signals and approximating the theoretical linear separability established in Theorem 1. We note that APF underperforms XGBGraph on Amazon and Elliptic, likely due to the tabular and highly heterogeneous node features that favor tree-based models (Grinsztajn et al., 2022; Tang et al., 2023), as further discussed in Appendix D. Overall, these strong results validate the effectiveness of our proposed anomaly-aware pre-training and granularity-adaptive fine-tuning framework.

**Mitigation of Homophily Disparity.** To investigate the ability of our model to mitigate the impact of local homophily disparity, we conduct a fine-grained performance analysis on abnormal nodes with varying degrees of local homophily. Specifically, we divide the abnormal nodes in the test set into four quartiles based on their local homophily scores and compute the model performance within each group against the remaining normal nodes.

The results are visualized in Figure 1, 3, 13 and 14. We observe that detection performance typically declines as local homophily decreases, highlighting the difficulty of identifying anomalies in heterophilic regions and the need for node-adaptive mechanisms. APF consistently achieves stronger performance across all homophily quartiles, demonstrating enhanced robustness to homophily disparity. Such mitigation mainly stems from our anomaly-aware pre-training and the adaptive fusion mechanism, allowing the model to tailor its decision boundary to each node's local structural context and mitigating performance degradation under local homophily disparity.

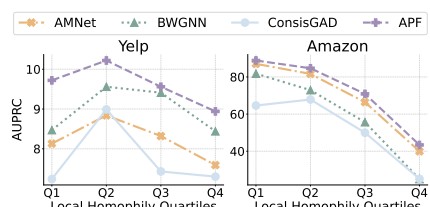

Figure 3: Performance variations across local homophily quartiles (Q1 = top 25%, Q4 = bottom 25%).

To provide a more fine-grained view, we further measure the performance variance across homophily levels by reporting the AUPRC differences between lower-homophily groups (Q2, Q3, Q4) and the highest-homophily group (Q1). The results in Table 2 show that APF generally yields the smallest or near-smallest performance variance across quartiles. This indicates that, although performance degradation remains as local homophily decreases, APF alleviates the disparity more effectively and yields more stable performance under challenging heterophilic conditions.

Table 2: AUPRC differences (Q1 - Q*) across local homophily quartiles. Smaller absolute values indicate lower performance gaps under homophily disparity.

| Dataset | Weibo | | | T-Finance | | | YelpChi | | | Amazon | | |
|---|---|---|---|---|---|---|---|---|---|---|---|---|
| | Q1-Q2 | Q1-Q3 | Q1-Q4 | Q1-Q2 | Q1-Q3 | Q1-Q4 | Q1-Q2 | Q1-Q3 | Q1-Q4 | Q1-Q2 | Q1-Q3 | Q1-Q4 |
| AMNet | *9.40* | *32.88* | *27.32* | *3.44* | 30.27 | **58.66** | *-0.71* | *-0.19* | 0.54 | 5.42 | 20.65 | 47.10 |
| BWGNN | 10.73 | 39.18 | 44.89 | 9.47 | 56.56 | 86.60 | -1.09 | -0.94 | **0.03** | 8.89 | 26.14 | 56.74 |
| ConsisGAD | 13.43 | 43.95 | 54.96 | 3.75 | *25.04* | 89.25 | -1.75 | *-0.19* | -0.06 | **-3.24** | **14.63** | **39.47** |
| **APF** | **-1.18** | **5.72** | **31.20** | **2.06** | **23.36** | *76.12* | **-0.50** | **0.16** | 0.78 | *4.18* | *17.98* | *45.46* |

## 4.3 MODEL ANALYSES

**Contribution of Each Component.** We perform a comprehensive ablation study to disentangle the contributions of the major components in both the pre-training and fine-tuning stages of APF. In the *pre-training* stage, we design four controlled variants: (i) using only the low-pass filter $g_L(\cdot)$, (ii) using only the high-pass filter $g_H(\cdot)$, (iii) removing the Rayleigh Quotient-guided subgraph $\mathcal{G}^{RQ}$ by applying standard DGI on $g_L(\cdot)$ and $g_H(\cdot)$, and (iv) replacing $\mathcal{G}^{RQ}$ with a full $k$-hop subgraph "✛". In the *fine-tuning* stage, we examine two factors: (i) discarding the anomaly-aware regularization loss $\mathcal{L}_{reg}$, and (ii) replacing our node- and dimension-adaptive fusion with mean pooling, concatenation, or attention-based fusion (Chai et al., 2022).

The results, shown in Table 3, reveal several important findings. (1) **Dual-filter necessity.** Leveraging both low- and high-pass filters *clearly outperforms* using either alone, confirming that semantic regularities and anomaly-indicative irregularities must be jointly captured for GAD. (2) **Power of $\mathcal{G}^{RQ}$.** The Rayleigh Quotient-guided subgraph yields *more precise anomaly discrimination* than the full $k$-hop variant "✛", highlighting the Rayleigh Quotient as an *effective label-free anomaly indicator*. (3) **Adaptive fusion advantage.** Our node- and dimension-adaptive fusion consistently surpasses the alternatives, showing the importance of exploiting frequency-selective signals while adapting to node-specific contexts. (4) **Regularization gains.** The anomaly-aware regularization loss $\mathcal{L}_{reg}$ provides *additional boosts*, aligning representations with class-level fusion bias and further strengthening anomaly detection.

Table 3: Ablation study on each component of our APF.

| $g_L$ | $g_H$ | $\mathcal{G}^{RQ}$ | $\mathcal{L}_{reg}$ | Fusion | Reddit | | YelpChi | | Questions | | DGraph-Fin | |
|---|---|---|---|---|---|---|---|---|---|---|---|---|
| | | | | | AUPRC | AUROC | AUPRC | AUROC | AUPRC | AUROC | AUPRC | AUROC |
| ✓ | ✓ | ✓ | ✓ | NDApt | **5.9±0.9** | **66.8±3.9** | **28.4±1.4** | **68.2±2.3** | **12.3±1.6** | **71.9±2.1** | **2.9±0.2** | **72.4±1.3** |
| ✓ | ✓ | ✗ | ✓ | NDApt | 5.3±0.6 | 66.1±2.0 | 27.1±1.6 | 67.6±1.4 | 10.9±1.8 | 71.2±2.5 | 2.6±0.1 | 71.1±0.5 |
| ✓ | ✓ | ✛ | ✓ | NDApt | 5.5±0.7 | 66.5±3.8 | 27.4±2.8 | 67.3±2.3 | 10.8±1.6 | 71.0±2.3 | 2.7±0.2 | 71.0±2.2 |
| ✓ | ✗ | ✗ | ✗ | - | 4.9±0.5 | 65.3±1.7 | 18.7±1.2 | 56.5±1.3 | 11.8±1.2 | 70.6±1.2 | 2.5±0.1 | 71.0±1.0 |
| ✗ | ✓ | ✓ | ✗ | - | 4.5±0.5 | 60.1±3.0 | 24.4±2.4 | 64.0±2.5 | 6.3±0.7 | 66.1±3.2 | 2.6±0.1 | 71.1±0.5 |
| ✓ | ✓ | ✓ | ✗ | NDApt | 5.3±0.5 | 64.3±1.7 | 27.5±1.6 | 67.4±2.3 | 11.1±1.6 | 71.2±2.7 | 2.8±0.1 | 71.7±0.4 |
| ✓ | ✓ | ✓ | ✗ | Mean | 5.1±0.6 | 63.5±2.0 | 27.6±1.6 | 67.2±2.1 | 10.8±1.7 | 71.0±2.4 | 2.8±0.1 | 71.6±0.4 |
| ✓ | ✓ | ✓ | ✗ | Concat | 5.1±0.6 | 64.6±2.7 | 27.3±1.9 | 67.3±2.2 | 11.1±1.5 | 68.5±3.2 | 2.8±0.0 | 71.5±0.2 |
| ✓ | ✓ | ✓ | ✗ | Atten. | 5.2±0.7 | 62.8±2.3 | 26.2±1.6 | 66.7±1.7 | 8.3±2.2 | 67.5±3.5 | 2.8±0.0 | 71.7±0.3 |

**Visualization of Fusion Coefficients.** To gain visual insights into our node- and dimension-adaptive fusion, we present heatmaps of the learned fusion coefficients $C$ in Figure 4. For clarity, we focus on the top 6 dimensions of $C$ for 3 randomly selected abnormal nodes $a_1, a_2, a_3$ and 3 randomly selected normal nodes $n_1, n_2, n_3$. The heatmaps reveal substantial variation across both nodes and dimensions, confirming the adaptiveness of $C$. For Amazon, T-Finance, and Tolokers, abnormal nodes generally have lower coefficients than normal ones, aligning with the intuition that anomalies should rely more on high-pass (anomaly-sensitive) features, while normal nodes benefit more from low-pass (semantic) representations. In contrast, Weibo shows similar coefficient distributions between classes, likely due to the smaller local homophily gap between the two classes. Moreover, the variability of $C$ across dimensions suggests that APF learns to assign different levels of node-wise and dimension-wise importance of information from both encoders, enabling more expressive fusion that captures subtle, localized anomaly-relevant patterns.

**Additional Analyses.** Due to space limitations, please kindly see the Appendix for extended discussions. Specifically, we report the model's time complexity in Appendix G, and present efficiency analyses regarding training time and memory overhead in Appendix I.3. The effectiveness of our

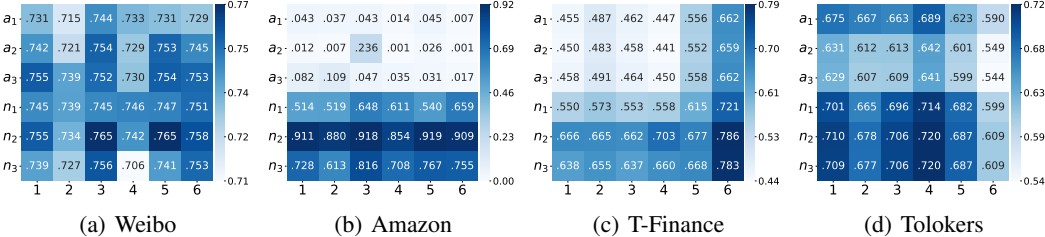

Figure 4: Visualization of the learned coefficients for the top 6 dimensions. The nodes $a_1, a_2, a_3$ are 3 randomly selected abnormal nodes, while $n_1, n_2, n_3$ are 3 randomly selected normal nodes.

pre-training under a purely unsupervised GAD setting is discussed in Appendix I.1. Moreover, Appendix I.2 provides visualizations of the learned low-pass, high-pass, and fused node representations. We also analyze the sensitivity to hyperparameters $p_a$ and $p_n$ in Appendix I.4, investigate performance under varying amounts of labeled nodes in Appendix I.5, and include additional figures and tables complementing the main results. We further re-classify the baselines into supervised, semi-supervised, and self-supervised categories in Appendix I.8, and compare our two-stage (pre-training and fine-tuning) approach against a joint learning strategy in Appendix I.9. These analyses jointly provide a more comprehensive understanding of the proposed framework.

## 5 CONCLUSION

This paper tackles label scarcity and homophily disparity in GAD by introducing APF. APF first leverages Rayleigh Quotient-guided subgraph sampling and dual spectral filters to capture both semantic and anomaly-sensitive signals without supervision. During fine-tuning, a node- and dimension-adaptive fusion mechanism, together with anomaly-aware regularization, enhances the model's ability to distinguish abnormal nodes under homophily disparity. Both theoretical analysis and extensive experiments on 10 benchmark datasets validate the effectiveness of our approach.

## ETHICS STATEMENT

We adhere to the ICLR Code of Ethics and have taken every measure to ensure that our research complies with the ethical guidelines set forth. Our work does not involve human subjects, nor does it raise any ethical concerns related to privacy or data usage. All datasets used in our experiments are publicly available, and we ensure that their release and use adhere to proper data-sharing policies. We have carefully selected the datasets and evaluation metrics to ensure fairness and transparency in our findings. No potential conflicts of interest, sponsorship biases, or ethical violations have been identified in our study. We commit to maintaining the highest standards of research integrity, and we are open to addressing any ethical concerns that may arise during the review process.

## REPRODUCIBILITY STATEMENT

We have made efforts to ensure that this work is fully reproducible. All experiments were conducted with well-defined and publicly available datasets, and the models were implemented using standard frameworks. To facilitate the reproducibility of our results, we provide a detailed description of our experimental setup in the main text and Appendix, including hyperparameters, training protocols, evaluation methods, and the open-source link. We believe that all materials necessary for reproducing our experiments are clearly outlined.

## ACKNOWLEDGMENTS

This work is partially supported by National Science and Technology Major Project (2026ZD16011200), the National Natural Science Foundation of China (62306137, 72371217), the Guangzhou Industrial Informatics and Intelligence Key Laboratory (2024A03J0628), the Nansha Key Area Science and Technology Project (2023ZD003, 2021JC02X191), and the Australian

Research Council under the streams of Future Fellowship (FT210100624), the Discovery Project (DP240101108, DP260100326), and the Linkage Project (LP230200892, LP240200546).

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

## A  USE OF LARGE LANGUAGE MODELS

In accordance with the ICLR 2026 Policies on Large Language Model Usage, we disclose that Large Language Models (LLMs) were employed during the preparation of this manuscript. Specifically, LLMs were used to (i) check grammar and spelling, (ii) improve clarity and conciseness of sentences, (iii) format tables for consistency, and (iv) shorten certain passages to reduce redundancy. Importantly, LLMs were not used to generate research ideas, design experiments, analyze results, or draft new scientific content. The authors remain fully responsible for all claims, findings, and conclusions presented in this work.

## B  DETAILED PRELIMINARIES

**Notations.**   In a general scenario, we are given an attributed graph $\mathcal{G} = (\mathcal{V}, \mathcal{E}, \boldsymbol{X})$, where $\mathcal{V} = \{v_1, \cdots, v_n\}$ is the set of $n$ nodes, $\mathcal{E} = \{e_{ij}\}$ is the set of edges, and $e_{ij} = (v_i, v_j)$ represents an edge between nodes $v_i$ and $v_j$. We define $\boldsymbol{A}$ as the corresponding adjacency matrix and $\boldsymbol{D}$ as the diagonal degree matrix with $\boldsymbol{D}_{ii} = d_i = \sum_j \boldsymbol{A}_{ij}$. The neighbor set $\mathcal{N}_i$ of each node $v_i$ is given by $\mathcal{N}_i = \{v_j : e_{ij} \in \mathcal{E}\}$. For each node $v_i$, it has a $d$-dimensional feature vector $\boldsymbol{x}_i \in \mathbb{R}^d$, and collectively the features of all nodes are denoted as $\boldsymbol{X} = (\boldsymbol{x}_1, \cdots, \boldsymbol{x}_n)^\top \in \mathbb{R}^{n \times d}$.

**Graph Anomaly Detection.**   GAD is formulated as a binary classification problem where anomalies are regarded as positive with label 1, while normal nodes are negative with label 0. Given $\mathcal{V}^L = \mathcal{V}_a \cup \mathcal{V}_n$, where $\mathcal{V}_a$ consists of labeled abnormal nodes and $\mathcal{V}_n$ comprises labeled normal nodes, along with their corresponding labels $\boldsymbol{y}^L$, the goal is to identify the anomalous status $\hat{\boldsymbol{y}}^U$ for the unlabeled nodes $\mathcal{V}^U = \mathcal{V} \setminus \mathcal{V}^L$. Usually, obtaining authentic labels is often costly, we assume that label information is only available for a small subset of nodes (i.e., $|\mathcal{V}^L| \ll |\mathcal{V}|$). Furthermore, there are usually significantly fewer abnormal nodes than normal nodes (i.e., $|\mathcal{V}_a| \ll |\mathcal{V}_n|$).

**Graph Spectral Filtering.**   Given the adjacency matrix $\boldsymbol{A}$, the graph Laplacian matrix is defined as $\boldsymbol{L} = \boldsymbol{D} - \boldsymbol{A}$, which is symmetric and positive semi-definite. Their symmetric normalized versions are noted as $\tilde{\boldsymbol{A}} = \boldsymbol{D}^{-1/2} \boldsymbol{A} \boldsymbol{D}^{-1/2}$ and $\tilde{\boldsymbol{L}} = \boldsymbol{I} - \boldsymbol{D}^{-1/2} \boldsymbol{A} \boldsymbol{D}^{-1/2}$. Its eigendecomposition is given by $\tilde{\boldsymbol{L}} = \boldsymbol{U} \boldsymbol{\Lambda} \boldsymbol{U}^\top$, where the columns of $\boldsymbol{U} \in \mathbb{R}^{n \times n}$ are orthonormal eigenvectors (graph Fourier basis), and $\boldsymbol{\Lambda} = \mathrm{diag}(\lambda_1, \lambda_2, \cdots, \lambda_n)$ contains the eigenvalues (frequencies), arranged such that $0 = \lambda_1 \le \lambda_2 \le \cdots \le \lambda_n$. Given the graph features $\boldsymbol{X}$ and a filter function $g(\cdot)$, the corresponding filtered features is thus defined as $\hat{\boldsymbol{X}} = \boldsymbol{U} g(\boldsymbol{\Lambda}) \boldsymbol{U}^\top \boldsymbol{X}$.

Typically, $\tilde{\boldsymbol{A}}$ acts as a low-pass filter with $g(\lambda) = 1 - \lambda$, while $-\tilde{\boldsymbol{A}}$ and $\tilde{\boldsymbol{L}}$ serve as high-pass filters, with $g(\lambda) = \lambda - 1$ and $g(\lambda) = \lambda$, respectively. In practice, a self-loop is often added to each node in the graph (i.e., $\boldsymbol{A} = \boldsymbol{A} + \boldsymbol{I}$) to alleviate numerical instabilities and improve performance Kipf & Welling (2017).

## C  RELATED WORKS

### C.1  GRAPH ANOMALY DETECTION

GNN-based approaches have emerged as a promising paradigm for GAD, due to their strong ability to capture both complex structural and node attribute patterns in graph data. A comprehensive and up-to-date survey of deep GAD methods is provided in (Qiao et al., 2025b), while BOND (Liu et al., 2022) and GADBench (Tang et al., 2023) establish performance benchmarks for unsupervised and semi-/supervised GAD approaches, respectively.

Unsupervised GAD approaches do not rely on labeled data for training and instead use unsupervised learning techniques to identify anomaly patterns in graph data. For instance, DOMINANT (Ding et al., 2019) employs a graph autoencoder to reconstruct both attributes and structure using GNNs. CoLA (Liu et al., 2021b) explores the consistency between anomalies and their neighbors across different contrastive views to assess node irregularity. VGOD (Huang et al., 2023) combines variance-based and attribute reconstruction models to detect anomalies in a balanced manner. TAM (Qiao & Pang, 2023) introduces local affinity as an anomaly measure, aiming to learn tailored node representations for GAD by maximizing the local affinity between nodes and their neighbors. GAD-EBM (Roy et al., 2023) evaluates the likelihood of normal and anomalous nodes to address GAD with an energy-based model. DiffGAD (Li et al., 2025) leverages diffusion sampling to infuse the latent space with discriminative content and introduces a content-preservation mechanism that retains valuable information across different scales for GAD.

Semi-/supervised GAD approaches assume that labels for some normal and abnormal nodes are available for training. They aim to assign labels by learning a decision boundary between normal and abnormal nodes. For example, BWGNN (Tang et al., 2022) uses a Beta graph wavelet to learn band-pass filters that capture anomaly signals, while DSGAD (Zheng et al., 2025a) extends BWGNN with dynamic wavelets and feature fusion. AMNet (Chai et al., 2022) employs a restricted Bernstein polynomial parameterization to approximate filters in multi-frequency groups. CARE-GNN (Dou et al., 2020), PCGNN (Liu et al., 2021a), and GHRN (Gao et al., 2023) adaptively prune inter-class edges based on neighbor distributions or the graph spectrum. PMP (Zhuo et al., 2024) introduces a partitioned message-passing mechanism to handle homophilic and heterophilic neighbors independently. To address settings with limited labeled data, CGenGA (Ma et al., 2024) proposes a diffusion-based graph generation method to synthesize additional training nodes, while ConsisGAD (Chen et al., 2024b) incorporates learnable data augmentation to utilize the abundance of unlabeled data for consistency training. GGAD (Qiao et al., 2024) introduces a novel semi-supervised framework using only labeled normal nodes. SpaceGNN Dong et al. (2025) combines space projection, distance-aware propagation, and ensemble mechanisms across multiple latent spaces to improve generalization. It is worth noting that some approaches can also address label scarcity by leveraging auxiliary data, such as cross-network meta-learning (Meta-GDN (Ding et al., 2021b)) or cross-domain adaptation (Commander (Ding et al., 2021a), ACT (Wang et al., 2023)). In contrast, our proposed APF targets the *single-graph* setting where no external auxiliary networks or source domains are available, relying instead on mining intrinsic anomaly signals from unlabeled nodes within the target graph.

Additionally, several works have explored GAD within the pre-training and fine-tuning paradigm. DCI (Wang et al., 2021) decouples representation learning and classification through a cluster-enhanced self-supervised learning task. Cheng et al. (2024) evaluates the performance of DGI (Veličković et al., 2019) and GraphMAE (Hou et al., 2022) for GAD, demonstrating the potential of leveraging graph pre-training to enhance GAD with limited supervision. However, most existing methods pre-train a uniform global low-pass filter (e.g., GCN (Kipf & Welling, 2017)) and then fine-tune a classifier on frozen node representations. The homophily disparity in GAD presents a significant challenge for directly applying these methods. To address this, we propose a pre-training and fine-tuning framework tailored for GAD.

In addition to the above work, PReNet (Pang et al., 2023) and NSReg (Wang et al., 2025) aim to identify anomalies whose patterns differ from labeled examples by enforcing strong normality modeling. GDN-AugAN (Zhou et al., 2023) enhances cross-dataset robustness through augmentation-based domain generalization. UniGAD (Lin et al., 2024), ARC (Liu et al., 2024a), UNPrompt (Niu et al., 2025), and AnomalyGFM (Qiao et al., 2025a) have explored foundation models for generalist GAD. While UniGAD (Lin et al., 2024) also employs a Rayleigh Quotient-based sampler, it uses the sampler as a unification tool to transform node/edge tasks into graph tasks for multi-level detection. In contrast, APF utilizes the sampler specifically to inject anomaly awareness during pre-training to address label scarcity in node anomaly detection.

## C.2 GRAPH PRE-TRAINING

Graph pre-training has emerged as a promising paradigm for label-efficient learning (Cheng et al., 2023; Ju et al., 2024). These methods first learn universal knowledge from unlabeled data using self-supervised objectives, which are then transferred to tackle specific downstream tasks. Existing

pre-training approaches can be broadly categorized into two groups: contrastive and non-contrastive approaches.

Contrastive approaches typically follow the principle of mutual information maximization (Hjelm et al., 2019), where the objective functions contrast positive pairs against negative ones. For instance, DGI (Veličković et al., 2019) and DCI (Wang et al., 2021) focus on representation learning by maximizing the mutual information between node-level representations and a global summary representation. GRACE (Zhu et al., 2020), GCA (Zhu et al., 2021), and NS4GC (Liu et al., 2024b) learn node representations by pulling together the representations of the same node (positive pairs) across two augmented views, while pushing apart the representations of different nodes (negative pairs) across both views.

Non-contrastive approaches, on the other hand, eliminate the need for negative samples. For example, CCA-SSG (Zhang et al., 2021) and G-BT (Bielak et al., 2022) aim to learn augmentation-invariant information while introducing feature decorrelation to capture orthogonal features. BGRL (Thakoor et al., 2022) and BLNN (Liu et al., 2024d) employ asymmetric architectures that learn node representations by predicting alternative augmentations of the input graph and maximizing the similarity between the predictions and their corresponding targets. GraphMAE Hou et al. (2022) focuses on feature reconstruction using a masking strategy and scaled cosine error. Additionally, SSGE (Liu et al., 2024c) minimizes the distance between the distribution of learned representations and the isotropic Gaussian distribution to promote the uniformity of node representations.

However, the methods discussed above rely on low-pass GNN encoders that inherently smooth neighbor representations, leading to unsatisfactory performance on heterophilic abnormal nodes. Although a recent work, PolyGCL (Chen et al., 2024a), employs both low- and high-pass encoders, it combines them using a simple linear strategy to obtain final node representations for fine-tuning. This approach is less flexible and effective than our proposed node- and dimension-adaptive fine-tuning strategy, as demonstrated in Theorem 1.

## D  LIMITATIONS

While APF demonstrates strong performance across a wide range of benchmarks, it exhibits limitations in certain scenarios. Specifically, APF underperforms tree-based models such as XGBGraph on datasets like Amazon and Elliptic, where node features are highly heterogeneous and dominate over structural signals. This suggests that in such cases, tree-based models, known for their robustness to feature heterogeneity, may outperform deep learning-based GNNs Grinsztajn et al. (2022). Future work could explore hybrid architectures that better integrate rich tabular features with graph topology.

Moreover, although APF is designed as a tailored framework for graph anomaly detection and holds promise in real-world applications such as financial fraud and cybersecurity, false positives remain a concern. Misclassifying normal nodes as anomalies may lead to unnecessary disruptions or adverse consequences for benign users. Addressing such risks requires further research into uncertainty quantification and trustworthy anomaly detection in graph settings.

## E  PROOF OF THEOREM 1

*Proof.* We extend the argument of Baranwal et al. (2021) from a single-pattern $CSBM$ to the mixed-pattern, degree-corrected, and class-imbalanced $ASBM$ in Definition 1. Throughout the proof, we make the following standard assumptions:

1. the graph size is relatively large with $\omega(d \log d) \leq n \leq \mathcal{O}(poly(d))$;

2. sparsity level obeys $p_1, q_1, p_2, q_2 = \omega(\log^2 n/n)$;

3. degree parameters are bounded and centered as $\theta_{\min} \leq \theta_i \leq \theta_{\max}$ with $\chi := \theta_{\max}/\theta_{\min} = \mathcal{O}(1)$ and $\mathbb{E}[\theta_i \mid z_i] = 1$ for $z_i \in \{a, n\}$;

4. class prior is imbalanced with $\pi_a = n_a/n \ll \pi_n = 1 - \pi_a$.

Following Baranwal et al. (2021); Ma et al. (2022); Mao et al. (2023); Han et al. (2024), we adopt the random-walk operator $S = D^{-1}A$ as a low-pass filter and its negative $-S$ as a high-pass filter. Concretely, each node $v_i$ is assigned a spectral filter function $g(\cdot)$ depending on its pattern: $g(\lambda) = \lambda$ if $v_i \in \mathcal{H}_o$ (homophilic), and $g(\lambda) = -\lambda$ if $v_i \in \mathcal{H}_e$ (heterophilic). This yields the node-adaptive filtered features $\hat{X} = U g(\Lambda) U^\top X$.

**Concentration under degree correction.** Let degree $d_i = \sum_j A_{ij}$ and recall that, conditional on $(z_i, h_i)$, we have $\mathbb{E}[A_{ij} \mid z_i, z_j, h_i] = \theta_i \theta_j \mathbf{B}^{(h_i)}_{z_i, z_j}$. Standard Bernstein (Chung & Lu, 2006; Lei & Rinaldo, 2015) bounds with bounded $\theta$ and $\omega(\log^2 n/n)$ sparsity yield

$$d_i = \theta_i \, n \big( \pi_a \beta^{(h_i)}_{z_i,a} + \pi_n \beta^{(h_i)}_{z_i,n} \big) (1 \pm o(1)),$$

where $\beta^{(h)}_{u,v} := \mathbf{B}^{(h)}_{u,v}$. Hence for any node $v_i$ and any unit vector $w$,

$$\hat{X}_i = \pm \frac{1}{d_i} \sum_j A_{ij} \, X_j = \pm \Big( \mathbb{E}[\hat{X}_i \mid z_i, h_i] + \xi_i \Big), \quad |\langle \xi_i, w \rangle| = \mathcal{O}\Bigg( \sqrt{\frac{\log n}{d \, d_i}} \Bigg) = \mathcal{O}\Bigg( \sqrt{\frac{\log n}{d \, n \, \theta_i \, \kappa_i}} \Bigg),$$

with $\kappa_i := \pi_a \beta^{(h_i)}_{z_i,a} + \pi_n \beta^{(h_i)}_{z_i,n}$. The sign $\pm$ corresponds to low- vs. high-pass filtering. Let the effective average connectivity under imbalance be

$$\kappa_{\mathrm{eff}} := \min \big\{ \pi_a p_1 + \pi_n q_1, \ \pi_a q_1 + \pi_n p_1, \ \pi_a p_2 + \pi_n q_2, \ \pi_a q_2 + \pi_n p_2 \big\}.$$

Using $\theta_i \geq \theta_{\min}$ and $\kappa_i \geq \kappa_{\mathrm{eff}}$, we obtain the uniform deviation bound (Baranwal et al., 2021)

$$\big| \langle \hat{X}_i - \mathbb{E}[\hat{X}_i \mid z_i, h_i], w \rangle \big| = \mathcal{O}\Bigg( \sqrt{\frac{\log n}{d \, n \, \kappa_{\mathrm{eff}}}} \Bigg), \tag{14}$$

matching the CSBM rate up to the effective factor $\kappa_{\mathrm{eff}}$.

**Pattern-dependent means after node-adaptive filtering.** Condition on $z_i$ and $h_i$. A neighboring node $v_j$ belongs to class $a$ with probability proportional to $\theta_j \beta^{(h_i)}_{z_i,a}$ and to class $n$ with probability proportional to $\theta_j \beta^{(h_i)}_{z_i,n}$. By (iii) and bounded heterogeneity $\chi = \mathcal{O}(1)$, the neighbor-class proportions concentrate at

$$\omega^{(h_i)}_{i,a} = \frac{\pi_a \beta^{(h_i)}_{z_i,a}}{\pi_a \beta^{(h_i)}_{z_i,a} + \pi_n \beta^{(h_i)}_{z_i,n}}, \quad \omega^{(h_i)}_{i,n} = 1 - \omega^{(h_i)}_{i,a}.$$

Thus the low-pass mean is a degree-normalized mixture

$$\mathbb{E}[SX]_i = \omega^{(h_i)}_{i,a} \mu + \omega^{(h_i)}_{i,n} \nu \, (1 \pm o(1)),$$

while the high-pass mean flips the sign: $\mathbb{E}[-SX]_i = -\mathbb{E}[SX]_i$. Enumerating cases gives, for $h_i = o$ (homophily):

$$\mathbb{E}[\hat{X}_i] = \begin{cases} \frac{\pi_a p_1 \mu + \pi_n q_1 \nu}{\pi_a p_1 + \pi_n q_1} (1 \pm o(1)), & z_i = a, \\ \frac{\pi_a q_1 \mu + \pi_n p_1 \nu}{\pi_a q_1 + \pi_n p_1} (1 \pm o(1)), & z_i = n, \end{cases}$$

and for $h_i = e$ (heterophily) the same expressions with $(p_1, q_1)$ replaced by $(p_2, q_2)$, followed by an overall sign flip due to the high-pass ($-S$). Consequently, under the node-adaptive choice (low-pass on $\mathcal{H}_o$ and high-pass on $\mathcal{H}_e$), all class-wise means align in the same ordering along direction $\nu - \mu$:

$$\langle \mathbb{E}[\hat{X}_i \mid z_i = a], \nu - \mu \rangle < \langle \mathbb{E}[\hat{X}_i \mid z_i = n], \nu - \mu \rangle,$$

with a gap proportional to $\Delta_h := \frac{|p_h - q_h|}{\pi_a p_h + \pi_n q_h + \pi_a q_h + \pi_n p_h}$ (here $h \in \{1, 2\}$ indexes $(p_1, q_1)$ or $(p_2, q_2)$), hence lower bounded by a constant multiple of $\frac{|p_h - q_h|}{p_h + q_h}$ and independent of $\theta$ due to the random-walk normalization.

**Prior-aware linear separator and margin.** Consider the linear classifier with direction $\boldsymbol{w}^* = R\frac{\boldsymbol{\nu} - \boldsymbol{\mu}}{\|\boldsymbol{\mu} - \boldsymbol{\nu}\|}$ and a bias that accounts for the class prior shift

$$b^* = -\frac{\langle \boldsymbol{\mu} + \boldsymbol{\nu}, \boldsymbol{w}^* \rangle}{2} + \tau_\pi, \qquad \tau_\pi = R\frac{\log(\pi_a/\pi_n)}{\|\boldsymbol{\mu} - \boldsymbol{\nu}\|}.$$

The additive term $\tau_\pi$ is the standard LDA correction under unequal priors with spherical covariances (Hastie et al., 2009). For any $v_i \in \mathcal{C}_a$, combining Step 2 and Eq 14 gives

$$\langle \hat{\boldsymbol{X}}_i, \boldsymbol{w}^* \rangle + b^* = \langle \mathbb{E}[\hat{\boldsymbol{X}}_i \mid z_i = a], \boldsymbol{w}^* \rangle - \frac{\langle \boldsymbol{\mu} + \boldsymbol{\nu}, \boldsymbol{w}^* \rangle}{2} + \tau_\pi + \underbrace{\langle \hat{\boldsymbol{X}}_i - \mathbb{E}[\hat{\boldsymbol{X}}_i], \boldsymbol{w}^* \rangle}_{\text{fluctuation}}$$

$$= -\frac{R}{2}\Gamma_{h_i}\|\boldsymbol{\mu} - \boldsymbol{\nu}\|(1 \pm o(1)) + \tau_\pi + \mathcal{O}\left(R\sqrt{\frac{\log n}{d\, n\, \kappa_{\text{eff}}}}\right),$$

where $\Gamma_{h_i} > 0$ depends on the pattern (homophily vs. heterophily) via the mixture coefficients in Step 2 (and is uniformly bounded away from 0 as $|p_h - q_h| > 0$). For $i \in \mathcal{C}_n$ the sign of the leading term is positive. Hence, if the feature center distance satisfies

$$\|\boldsymbol{\mu} - \boldsymbol{\nu}\| \geq C\frac{\log n}{\sqrt{d\, n\, \kappa_{\text{eff}}}}$$

for a sufficiently large constant $C > 0$, the margin term dominates both the stochastic fluctuation $\mathcal{O}\left(R\sqrt{\frac{\log n}{d\, n\, \kappa_{\text{eff}}}}\right)$ and the prior correction $\tau_\pi = \mathcal{O}\left(R|\log(\pi_a/\pi_n)|/\|\boldsymbol{\mu} - \boldsymbol{\nu}\|\right)$ even when $\pi_a \ll \pi_n$. Therefore, according to part 2 of Theorem 1 in Baranwal et al. (2021), $\text{sign}(\langle \hat{\boldsymbol{X}}_i, \boldsymbol{w}^* \rangle + b^*)$ equals the true label for all nodes with probability $1 - o_d(1)$, by a union bound over $v_i \in \mathcal{V}$. This establishes linear separability of node-adaptively filtered features under degree correction and class imbalance, proving the theorem. $\square$

## F    FORMULATIONS OF FUSION METHODS

Here, we provide the mathematical formulations of the fusion methods described in Section 4.3. These fusion methods aim to generate overall node representations $\boldsymbol{Z}$ by combining the representations generated by the low-pass encoder ($\boldsymbol{Z}_L \in \mathbb{R}^{n \times e}$) and high-pass encoder ($\boldsymbol{Z}_H \in \mathbb{R}^{n \times e}$).

- The "Mean" method averages the representations from the low-pass and high-pass encoders, i.e., $\boldsymbol{Z} = 0.5 \cdot (\boldsymbol{Z}_L + \boldsymbol{Z}_H)$.
- The "Concat" method concatenates the representations from the low-pass and high-pass encoders, i.e., $\boldsymbol{Z} = [\boldsymbol{Z}_L, \boldsymbol{Z}_H]$.
- The "Atten." method Chai et al. (2022) employs an attention mechanism to learn the weights $\boldsymbol{\alpha}_L, \boldsymbol{\alpha}_H \in [0, 1]^{n \times 1}$ for $n$ nodes, such that $\boldsymbol{Z} = \boldsymbol{\alpha}_L \cdot \boldsymbol{Z}_L + \boldsymbol{\alpha}_H \cdot \boldsymbol{Z}_H$. Specifically, for node $v_i$ with $\boldsymbol{Z}_i^L, \boldsymbol{Z}_i^H \in \mathbb{R}^{1 \times e}$, the attention scores are computed as:

$$\begin{aligned} \omega_i^L &= \boldsymbol{q}^\top \cdot \tanh\left(\boldsymbol{W}_Z^L \boldsymbol{Z}_i^{L^\top} + \boldsymbol{W}_X^L \boldsymbol{x}_i\right), \\ \omega_i^H &= \boldsymbol{q}^\top \cdot \tanh\left(\boldsymbol{W}_Z^H \boldsymbol{Z}_i^{H^\top} + \boldsymbol{W}_X^H \boldsymbol{x}_i\right), \end{aligned} \tag{15}$$

  where $\boldsymbol{W}_Z^L, \boldsymbol{W}_Z^H \in \mathbb{R}^{e' \times e}$ and $\boldsymbol{W}_X^L, \boldsymbol{W}_X^H \in \mathbb{R}^{e' \times d}$ are learnable parameter matrices, and $\boldsymbol{q} \in \mathbb{R}^{e' \times 1}$ is the shared attention vector. The final attention weights of node $v_i$ are obtained by normalizing the attention values using the softmax function:

$$\begin{aligned} \alpha_i^L &= \frac{\exp(\omega_i^L)}{\exp(\omega_i^L) + \exp(\omega_i^H)}, \\ \alpha_i^H &= \frac{\exp(\omega_i^H)}{\exp(\omega_i^L) + \exp(\omega_i^H)}. \end{aligned} \tag{16}$$

## G    TIME COMPLEXITY

We analyze the time complexity of our proposed APF framework by dividing the computation into three stages. Let $n$ and $m$ denote the number of nodes and edges in the graph, respectively, and let $K$ be the order of the spectral polynomial filters.

**Preprocessing.** We first extract a Rayleigh Quotient-guided subgraph $\mathcal{G}_i^{RQ}$ for each node using the MRQSampler Lin et al. (2024), which has a total complexity of $O(n \log n)$. Since the sampling for each node is independent, this step can be parallelized to further reduce runtime. Importantly, this subgraph sampling is performed only once and reused in both training and inference, thereby introducing minimal overhead.

**Pre-training.** The coefficients of the polynomial filters in APF can be precomputed in time linear to $K$. Given that $K$-order spectral filters propagate information across $K$ hops, the filtering process requires $O(Km + Kn)$ time. The loss function $\mathcal{L}_{pt}$, which operates over all nodes and edges, incurs an additional $O(n + m)$ cost. The overall time complexity of the pre-training stage is therefore $O((K + 1)(m + n))$, which scales linearly with the graph size and filter order.

**Fine-tuning.** The gated fusion network computes adaptive coefficients with complexity $O(n)$, and the two-layer MLP used for classification adds another $O(2n)$. The anomaly-aware loss $\mathcal{L}_{ft}$ involves only the labeled nodes and thus contributes $O(l)$, where $l$ is the number of labeled nodes. The total fine-tuning complexity is $O(3n + l)$.

In summary, the pre-training and fine-tuning phases of APF scale linearly with the graph size. Given that the subgraph extraction is only performed once and can be computed in parallel, APF exhibits strong scalability and is well-suited for large-scale graphs. For example, our model can be applied to datasets like T-Social, which contains over 5.7 million nodes and 73 million edges. We additionally conduct efficiency comparison in Appendix I.3.

# H  ADDITIONAL EXPERIMENTAL DETAILS

## H.1  DATASETS

Following GADBench Tang et al. (2023), we conduct experiments on 10 real-world datasets spanning various scales and domains. Reddit Kumar et al. (2019), Weibo Kumar et al. (2019), Questions Platonov et al. (2023), and T-Social Tang et al. (2022) focus on detecting anomalous accounts on social media platforms. Tolokers Platonov et al. (2023), Amazon McAuley & Leskovec (2013), and YelpChi Rayana & Akoglu (2015) are designed to identify fraudulent workers, reviews, and reviewers in crowdsourcing or e-commerce platforms. T-Finance Tang et al. (2022), Elliptic Weber et al. (2019), and DGraph-Fin Huang et al. (2022) target the detection of fraudulent users, illicit entities, and overdue loans in financial networks. Dataset Statistics are presented in Table 4. Detailed descriptions of these datasets are as follows.

- **Reddit** Kumar et al. (2019): This dataset includes a user-subreddit graph, capturing one month's worth of posts shared across various subreddits. It includes verified labels for banned users and focuses on the 1,000 most active subreddits and the 10,000 most engaged users, resulting in 672,447 interactions. Posts are represented as feature vectors based on Linguistic Inquiry and Word Count (LIWC) categories.

- **Weibo** Kumar et al. (2019): This dataset consists of a user-hashtag graph from the Tencent-Weibo platform, containing 8,405 users and 61,964 hashtags. Suspicious activities are defined as posting two messages within a specific time frame, such as 60 seconds. Users engaged in at least five such activities are labeled as "suspicious," while others are categorized as "benign." The feature vectors are based on the location of posts and bag-of-words features.

- **Amazon** McAuley & Leskovec (2013): This dataset focuses on detecting users who are paid to write fake reviews for products in the Musical Instrument category on Amazon.com. It contains three types of relationships: U-P-U (users reviewing the same product), U-S-U (users giving the same star rating within one week), and U-V-U (users with top 5% mutual review similarities).

- **YelpChi** Rayana & Akoglu (2015): This dataset aims to identify anomalous reviews on Yelp.com that unfairly promote or demote products or businesses. It includes three types of edges: R-U-R (reviews by the same user), R-S-R (reviews for the same product with the same star rating), and R-T-R (reviews for the same product within the same month).

Table 4: Dataset statistics including the number of nodes and edges, the node feature dimension, the ratio of anomalies, the local homophily, the concept of relations, and the type of node features. $h^a$, and $h^n$ represent the average local homophily for abnormal nodes and normal nodes respectively. *As shown, $h^a$ is much smaller than $h^n$, highlighting the presence of class-level local homophily disparity.* "Misc." refers to node features that are a combination of heterogeneous attributes, which may include categorical, numerical, and temporal information.

| Dataset | #Nodes | #Edges | #Feat. | Anomaly | $h^a$ | $h^n$ | Relation Concept | Feature Type |
|---------|--------|--------|--------|---------|-------|-------|------------------|--------------|
| Reddit | 10,984 | 168,016 | 64 | 3.3% | 0.000 | 0.994 | Under Same Post | Text Embedding |
| Weibo | 8,405 | 407,963 | 400 | 10.3% | 0.858 | 0.977 | Under Same Hashtag | Text Embedding |
| Amazon | 11,944 | 4,398,392 | 25 | 9.5% | 0.102 | 0.968 | Review Correlation | Misc. Information |
| YelpChi | 45,954 | 3,846,979 | 32 | 14.5% | 0.195 | 0.867 | Reviewer Interaction | Misc. Information |
| T-Finance | 39,357 | 21,222,543 | 10 | 4.6% | 0.543 | 0.976 | Transaction Record | Misc. Information |
| Elliptic | 203,769 | 234,355 | 166 | 9.8% | 0.234 | 0.985 | Payment Flow | Misc. Information |
| Tolokers | 11,758 | 519,000 | 10 | 21.8% | 0.476 | 0.679 | Work Collaboration | Misc. Information |
| Questions | 48,921 | 153,540 | 301 | 3.0% | 0.111 | 0.922 | Question Answering | Text Embedding |
| DGraph-Fin | 3,700,550 | 4,300,999 | 17 | 1.3% | 0.013 | 0.997 | Loan Guarantor | Misc. Information |
| T-Social | 5,781,065 | 73,105,508 | 10 | 3.0% | 0.174 | 0.900 | Social Friendship | Misc. Information |

- **T-Finance** Tang et al. (2022): This dataset is designed to detect anomalous accounts in transaction networks. The nodes represent unique anonymized accounts with 10-dimensional features related to registration days, login activities, and interaction frequency. Edges represent transactions between accounts, and anomalies are annotated by human experts based on categories such as fraud, money laundering, and online gambling.

- **Elliptic** Weber et al. (2019): This dataset includes over 200,000 Bitcoin transactions (nodes), 234,000 directed payment flows (edges), and 166 node features. It maps Bitcoin transactions to real-world entities, categorizing them as either licit (e.g., exchanges, wallet providers, miners) or illicit (e.g., scams, malware, terrorist organizations, ransomware, and Ponzi schemes).

- **Tolokers** Platonov et al. (2023): This dataset is derived from the Toloka crowdsourcing platform. Nodes represent workers who have participated in at least one of 13 selected projects. An edge connects two workers if they collaborate on the same task. The goal is to predict which workers were banned in any of the projects. Node features are based on worker profiles and task performance.

- **Questions** Platonov et al. (2023): This dataset is collected from the Yandex Q question-answering platform. It includes users as nodes, with edges representing answers between users during a one-year period (September 2021 to August 2022). It focuses on users interested in the "medicine" topic. The task is to predict which users remained active by the end of the period. Node features include the mean of FastText embeddings for words in the user descriptions, with a binary feature indicating users without descriptions.

- **DGraph-Fin** Huang et al. (2022): This dataset is a large-scale dynamic graph from the Finvolution Group representing a financial industry social network. Nodes represent users, and edges indicate emergency contact relationships. Anomalous nodes correspond to users exhibiting overdue behaviors. The dataset includes over 3 million nodes, 4 million dynamic edges, and more than 1 million unbalanced ground-truth anomalies.

- **T-Social** Tang et al. (2022): This dataset targets anomalous accounts in social networks. Nodes share the same annotations and features as those in T-Finance, with edges representing friend relationships maintained for more than three months. Anomalous nodes are annotated by experts in categories like fraud, money laundering, and online gambling.

## H.2 BASELINES

We compare our model with a series of baseline methods, which can be categorized into the following groups: (1) Standard GNN Architectures, including GCN Kipf & Welling (2017), GIN Xu et al. (2019), GAT Veličković et al. (2018), ACM Luan et al. (2022), FAGCN Bo et al. (2021), AdaGNN Dong et al. (2021), and BernNet He et al. (2021); (2) GNNs Specialized for GAD, including GAS Li et al. (2019), DCI Wang et al. (2021), PCGNN Liu et al. (2021a), AMNet Chai et al. (2022), BWGNN Tang et al. (2022), GHRN Gao et al. (2023), ConsisGAD Chen et al. (2024b),

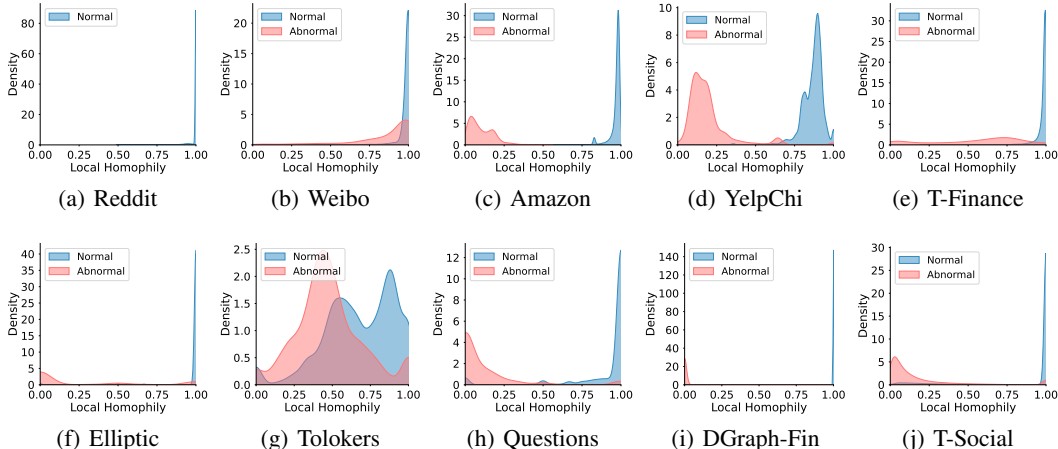

Figure 5: Distribution of local homophily across different datasets. GAD graphs display two levels of local homophily disparity: (1) Different nodes exhibit varying degrees of local homophily (node-level) and (2) Abnormal nodes tend to show lower local homophily than normal nodes (class-level). On Reddit, we only plot the distribution of local homophily for normal nodes, since the local homophily of all abnormal nodes is 0.

SpaceGNN Dong et al. (2025), and XGBGraph Tang et al. (2023); (3) Graph Pre-Training Methods, including DGI Veličković et al. (2019), GRACE Zhu et al. (2020), G-BT Bielak et al. (2022), Graph-MAE Hou et al. (2022), BGRL Thakoor et al. (2022), SSGE Liu et al. (2024c), PolyGCL Chen et al. (2024a), and BWDGI which incorporates BWGNN and DGI. Detailed descriptions of these baselines are as follows.

### H.2.1 STANDARD GNN ARCHITECTURES

- **GCN** Kipf & Welling (2017) employs a convolutional operation on the graph to propagate information from each node to its neighboring nodes, enabling the network to learn a representation for each node based on its local neighborhood.

- **GIN** Xu et al. (2019) is designed to capture the structural properties of a graph while preserving graph isomorphism. Specifically, it generates identical embeddings for graphs that are structurally identical, regardless of permutations in node labels.

- **GAT** Veličković et al. (2018) incorporates an attention mechanism, assigning different levels of importance to nodes during the information aggregation process. This allows the model to focus on the most relevant nodes within a neighborhood.

- **ACM** Luan et al. (2022) leverages low-, high, and full-pass spectral filters and an attention-based mixing mechanism to adaptively extract richer localized information for diverse node heterophily situations.

- **FAGCN** Bo et al. (2021) adaptively integrates low-frequency and high-frequency signals through a self-gating mechanism. This approach enhances the model's ability to handle both homophilic and heterophilic networks.

- **AdaGNN** Dong et al. (2021) leverages an adaptive frequency response filter to capture the varying importance of different frequency components for node representation learning. This approach improves the expressiveness of the model and alleviates the over-smoothing problem.

- **BernNet** He et al. (2021) provides a robust framework for designing and learning arbitrary graph spectral filters. It uses an order-K Bernstein polynomial approximation to estimate filters over the normalized Laplacian spectrum of a graph.

### H.2.2 GNNs SPECIALIZED FOR GAD

- **GAS** Li et al. (2019) is a highly scalable method for detecting spam reviews. It extends GCN to handle heterogeneous and heterophilic graphs and adapts to the graph structure of specific GAD applications using the KNN algorithm.

- **DCI** Wang et al. (2021) reduces inconsistencies between node behavior patterns and label semantics, and captures intrinsic graph properties within concentrated feature spaces by clustering the graph into multiple segments.

- **PCGNN** Liu et al. (2021a) uses a label-balanced sampler to select nodes and edges for training, ensuring a balanced label distribution in the induced subgraph. Additionally, it employs a learnable, parameterized distance function to select neighbors, filtering out redundant links while adding beneficial ones for improved fraud prediction.

- **AMNet** Chai et al. (2022) captures both low- and high-frequency signals by stacking multiple BernNets, adaptively combining signals from different frequency ranges.

- **BWGNN** Tang et al. (2022) addresses the "right-shift" phenomenon in graph anomalies, where spectral energy distribution shifts from low to high frequencies. It uses a Beta kernel to address high-frequency anomalies through flexible, spatially- and spectrally-localized band-pass filters.

- **GHRN** Gao et al. (2023) targets the heterophily problem in the spectral domain for graph anomaly detection. This method prunes inter-class edges to highlight and delineate the graph's high-frequency components.

- **ConsisGAD** Chen et al. (2024b) focuses on graph anomaly detection with limited supervision. It incorporates learnable data augmentation to utilize the abundance of unlabeled data for consistency training.

- **SpaceGNN** Dong et al. (2025) integrates learnable space projection, distance-aware propagation, and multiple space ensemble modules to leverage the benefits of different spaces (Euclidean, hyperbolic, and spherical) for node anomaly detection with extremely limited labels.

- **XGBGraph** Tang et al. (2023) first aggregates features from neighboring nodes to enhance the representation of each node, and then uses XGBoost Chen & Guestrin (2016) to classify nodes as normal or anomalous. This approach leverages the robustness and efficiency of tree ensembles while incorporating graph structure to improve anomaly detection performance.

### H.2.3 GRAPH PRE-TRAINING METHODS

- **DGI** Veličković et al. (2019) learns representations by maximizing the mutual information between node representations and a global summary representation.

- **GRACE** Veličković et al. (2019) learns node representations by pulling together the representations of the same node (positive pairs) across two augmented views, while pushing apart the representations of different nodes (negative pairs) across both views.

- **GraphMAE** Hou et al. (2022) is a masked graph auto-encoder that focuses on feature reconstruction using both a masking strategy and scaled cosine error.

- **BGRL** Thakoor et al. (2022) employs asymmetric architectures to learn node representations by predicting alternative augmentations of the input graph and maximizing the similarity between these predictions and their corresponding targets.

- **G-BT** Bielak et al. (2022) utilizes a cross-correlation-based loss function to reduce redundancy in the learned representations, which enjoys fewer hyperparameters and significantly reduced computation time.

- **SSGE** Liu et al. (2024c) minimizes the distance between the distribution of learned representations and an isotropic Gaussian distribution, promoting the uniformity of node representations.

- **PolyGCL** Chen et al. (2024a) addresses heterophilic challenges in graph pre-training by using polynomial filters as encoders and incorporating a combined linear objective between low- and high-frequency components in the spectral domain.

- **BWDGI** pre-trains the state-of-the-art GAD backbone BWGNN Tang et al. (2022) using DGI Veličković et al. (2019) as the pretext objective.

### H.3 Evaluation Protocols

Following GADBench Tang et al. (2023), we evaluate performance using three popular metrics: the Area Under the Receiver Operating Characteristic Curve (AUROC), the Area Under the Precision-Recall Curve (AUPRC) calculated via average precision, and the Recall score within the top-$K$ predictions (Rec@K). Here, $K$ corresponds to the number of anomalies in the test set. We prioritize these threshold-independent (AUROC, AUPRC) and rank-based (Rec@K) metrics over the F1 score to avoid the instability associated with selecting decision thresholds under label-scarce conditions. For all metrics, anomalies are treated as the positive class, with higher scores indicating better model performance. To closely simulate real-world scenarios with limited supervision, we strictly adhere to the **semi-supervised setting** defined in GADBench. Accordingly, we standardize the training/validation set across all datasets to include 100 labels — **20** positive (abnormal) and 80 negative (normal) labels Tang et al. (2023). This specific configuration ensures our results are directly comparable to the semi-supervised benchmarks reported in the GADBench Appendix (Table 11). To ensure the robustness of our findings, we perform 10 random splits, as provided by GADBench, on each dataset and report the average performance.

### H.4 Implementation Details

We use the implementations of all baseline methods provided by GADBench Tang et al. (2023) or the respective authors. Our APF model is implemented using PyTorch and the Deep Graph Library (DGL) Wang et al. (2019). Experiments are conducted on a Linux server equipped with an Intel(R) Xeon(R) Gold 6248 CPU @ 2.50GHz and a 32GB NVIDIA Tesla V100 GPU.

During the pre-training phase, each model is trained for up to 800 epochs using the Adam optimizer Kingma & Ba (2015), with a patience of 20. Hyperparameters are tuned as follows: filter order $\in \{2, 3\}$, learning rate $\in \{0.01, 0.001, 0.0001\}$, representation dimension $\in \{32, 64\}$, activation function $\in \{\text{ReLU}, \text{ELU}, \text{PReLU}, \text{Tanh}\}$, and normalization $\in \{\text{none}, \text{batch}, \text{layer}\}$. For efficiency, we extract 1-hop subgraphs instead of 2-hop ones on denser or larger datasets Amazon, T-Finance, and T-Social.

During the fine-tuning phase, a 2-layer MLP classifier is trained for up to 500 epochs using the Adam optimizer Kingma & Ba (2015), with a learning rate of 0.01 and weight decay selected from $\{0.0, 0.01, 0.0001\}$. The classifier with the highest validation AUROC score is selected for testing. Regarding the regularization hyperparameters, while we conduct a full grid search ($p_n, p_a \in [0, 1]$) for the sensitivity analysis, we adopt an efficient strategy for practical tuning: we fix $p_n$ to a high default value (e.g., 0.9 or 1.0) and perform a small search for $p_a$ (e.g., $\{0.0, 0.1, 0.2, 0.3, 0.4\}$), subject to $p_a \leq p_n$. Our implementation codes are available at `https://github.com/Cloudy1225/APF`.

## I Additional Experiments

### I.1 Pretraining-only for Unsupervised GAD

To further verify the effectiveness of our pre-training, we also investigate its performance under a purely unsupervised scenario where no anomaly labels are available. In this case, the pre-training stage remains unchanged. After obtaining the pre-trained low- and high-pass node representations, we directly concatenate them and feed the representations into Isolation Forest (IF) (Liu et al., 2008), a widely used ensemble method for unsupervised anomaly detection, to derive anomaly scores for each node. This modification enables our framework to operate in a label-free manner, aligning with common practice in unsupervised GAD.

We follow the evaluation pipeline of the unsupervised GAD benchmark BOND (Liu et al., 2022), and select the three datasets overlapping with ours: Reddit, Weibo, and DGraph. We include comparisons with two recent state-of-the-art unsupervised baselines, GAD-EBM (Roy et al., 2023)

and DiffGAD (Li et al., 2025). Table 5 reports AUROC results, where baseline numbers are taken from the respective papers.

Table 5: Unsupervised GAD results (AUROC %).

| Method | Reddit | Weibo | DGraph |
|---|---|---|---|
| GAD-EBM (Roy et al., 2023) | 58.5±1.6 | 93.2±1.8 | 60.3±2.5 |
| DiffGAD (Li et al., 2025) | 56.3±0.1 | **93.4±0.3** | 52.4±0.0 |
| IF (raw feats) (Liu et al., 2008) | 45.2±1.7 | 53.5±2.8 | 60.9±0.7 |
| **IF + Our Pre-training** | **59.9±2.4** | 93.1±1.8 | **63.3±0.6** |
| Ours (Semi-supervised) | 66.8±3.9 | 98.8±0.3 | 72.4±1.3 |

From the results, we observe that: (i) pretraining substantially improves the performance of IF, which alone struggles to capture structural anomalies; (ii) our method surpasses the state-of-the-art DiffGAD on Reddit and DGraph in the unsupervised setting; (iii) nevertheless, the semi-supervised version of our full framework still achieves the best performance, indicating that anomaly-aware pretraining brings general benefits while labels further boost detection accuracy.

Overall, these findings demonstrate that the proposed pre-training not only benefits semi-supervised settings but also provides clear gains in purely unsupervised anomaly detection, further validating its general effectiveness.

## I.2 VISUALIZATION OF LEARNED REPRESENTATIONS

To provide more intuitive insights into the learned representations, we apply t-SNE to visualize the distributions of $Z_L$, $Z_H$, and their fused representation $Z$. As shown in Figure 6, the results consistently reveal clear patterns: (i) the low-pass representation $Z_L$ and the high-pass representation $Z_H$ typically occupy distinct and largely non-overlapping regions in the embedding space, indicating that they capture complementary signals; (ii) the fused representation $Z$ exhibits partial overlaps with both $Z_L$ and $Z_H$, suggesting that it effectively integrates information from both frequency domains. These visualizations thus provide direct evidence that the dual-filter encoding indeed extracts complementary information, and the adaptive fusion mechanism successfully combines them into more comprehensive node representations.

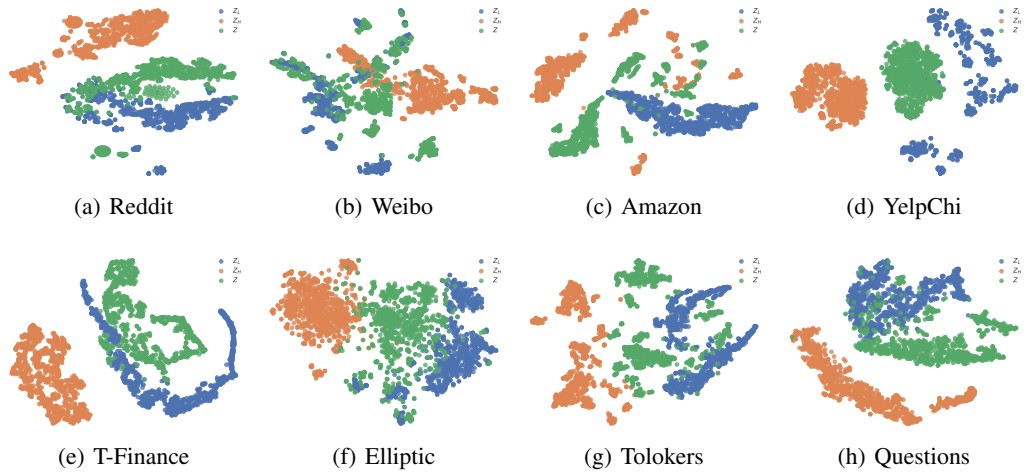

(a) Reddit     (b) Weibo     (c) Amazon     (d) YelpChi

(e) T-Finance     (f) Elliptic     (g) Tolokers     (h) Questions

Figure 6: Visualization of the learned representations $Z_L$, $Z_H$, and $Z$.

## I.3 EFFICIENCY COMPARISON.

We evaluate the training time and memory usage of APF on two large-scale datasets, YelpChi and T-Social, as shown in Table 6. Compared to end-to-end models like GCN, AMNet, and BWGNN, APF

incurs higher computational costs due to its pre-training phase. However, it remains more efficient than GHRN, which performs fine-grained edge-level operations and thus consumes substantially more memory, particularly on large graphs. When compared to models specifically designed for label-scarce settings, e.g., ConsisGAD and SpaceGNN, our method demonstrates lower training time and comparable or even lower memory usage. Overall, although APF introduces moderate computational overhead due to its two-stage design, the additional cost is justified by the significant gains in anomaly detection performance. These results demonstrate that APF is a viable and scalable solution for real-world, large-scale GAD applications.

Table 6: Efficiency comparison in terms of training time and GPU memory.

| Model | YelpChi | | T-Social | |
|---|---|---|---|---|
| | Time (s) | Mem. (MB) | Time (s) | Mem. (MB) |
| GCN | 1.93 | 547.73 | 61.75 | 13241.26 |
| AMNet | 4.09 | 773.38 | 213.82 | 18970.04 |
| BWGNN | 2.66 | 729.17 | 112.01 | 16146.46 |
| GHRN | 3.22 | 3360.71 | 161.00 | 28080.33 |
| ConsisGAD | 89.28 | 16390.42 | 1674.87 | 14145.61 |
| SpaceGNN | 13.80 | 24217.29 | 1273.43 | 20518.91 |
| APF | 7.89 | 1370.62 | 569.35 | 26351.84 |

### I.4 HYPERPARAMETER ANALYSIS

In addition to the adaptive fusion mechanism, our APF further introduces two hyperparameters, $p_a$ and $p_n$, which represent the expected preference for low-pass representations in abnormal and normal nodes, respectively. To assess their impact on our performance, we vary these hyperparameters from $0.0$ to $1.0$ in increments of $0.1$. The results are presented in Figure 7, 8 and 9. It is observed that the right half of the heatmap, corresponding to relatively larger $p_n$ values, generally outperforms the left half. This aligns with the understanding that normal nodes benefit more from low-pass representations for generic knowledge, due to their strong structural consistency with neighbors. Additionally, the optimal combination of $(p_a, p_n)$ always appears in the lower-right half of the heatmap, where $p_a \leq p_n$. This indicates that abnormal nodes are assigned lower $p_a$ values, thus placing greater emphasis on anomaly-indicative components, which better capture their deviation from the local context. These observations are consistent with the intuition behind our adaptive fusion design: normal and abnormal nodes require different emphases of knowledge to maximize discriminability. Overall, $p_a$ and $p_n$ provide a simple yet effective way that consistently guides APF toward strong and stable anomaly detection performance with minimal tuning effort.

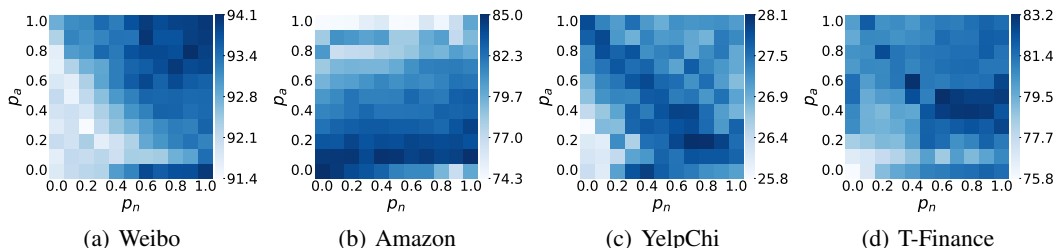

(a) Weibo          (b) Amazon          (c) YelpChi          (d) T-Finance

Figure 7: How the AUPRC score varies with different values of $p_a$ and $p_n$.

### I.5 UNDER VARYING SUPERVISION

This section evaluates the performance of APF under varying levels of supervision by modifying the number of labeled abnormal nodes. Following Tang et al. (2023), the number of labeled normal nodes is set to four times the number of labeled abnormal nodes. We present the results in terms of AUPRC, AUROC, and Rec@K in Figures 10, 11, and 12, respectively. As expected, performance generally improves across all methods as the number of labeled nodes increases. Notably, APF delivers consistent improvements over baseline pre-training methods and surpasses the state-of-the-art GAD

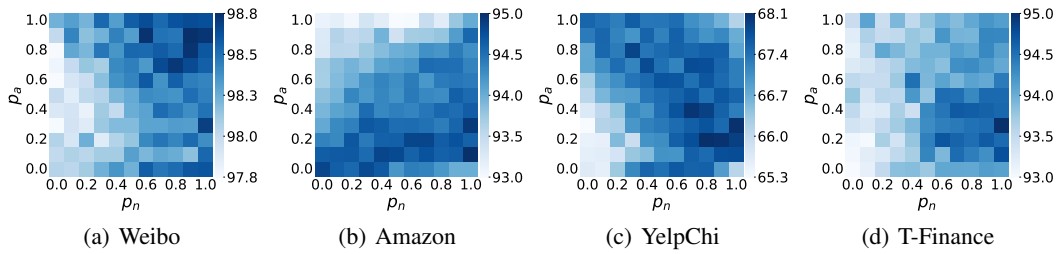

(a) Weibo          (b) Amazon          (c) YelpChi          (d) T-Finance

Figure 8: How the AUROC score of APF varies with different values of $p_a$ and $p_n$.

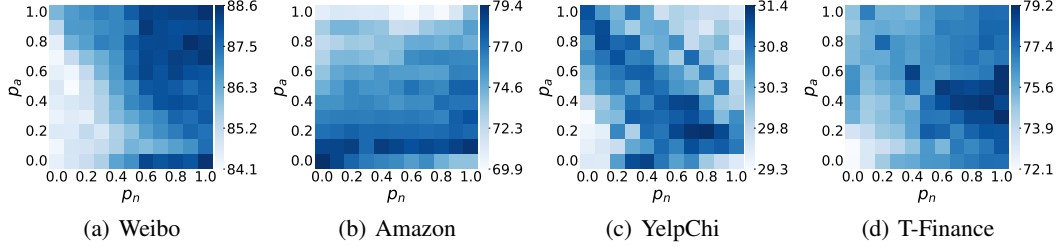

(a) Weibo          (b) Amazon          (c) YelpChi          (d) T-Finance

Figure 9: How the Rec@K score of APF varies with different values of $p_a$ and $p_n$.

models, even with only 5 labeled abnormal nodes. This highlights the effectiveness of our approach in addressing GAD with limited supervision.

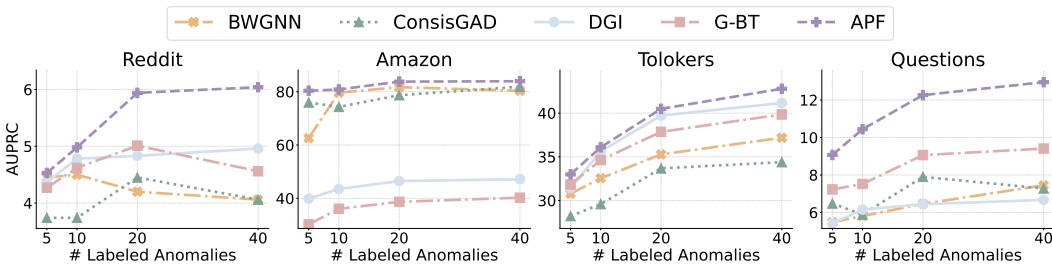

Figure 10: How the AUPRC score varies with different numbers of labeled anomalies.

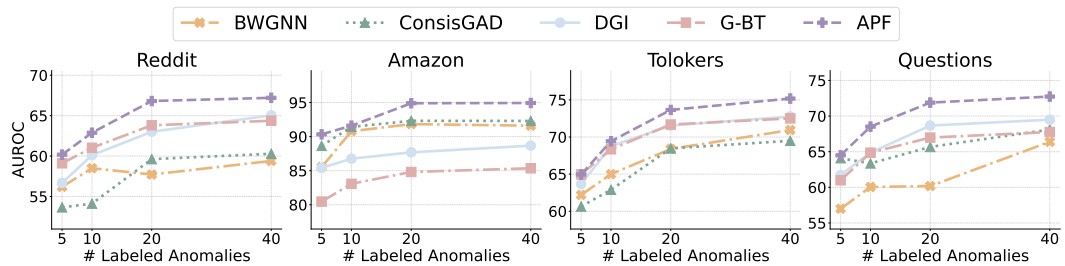

Figure 11: How the AUROC score varies with different numbers of labeled anomalies.

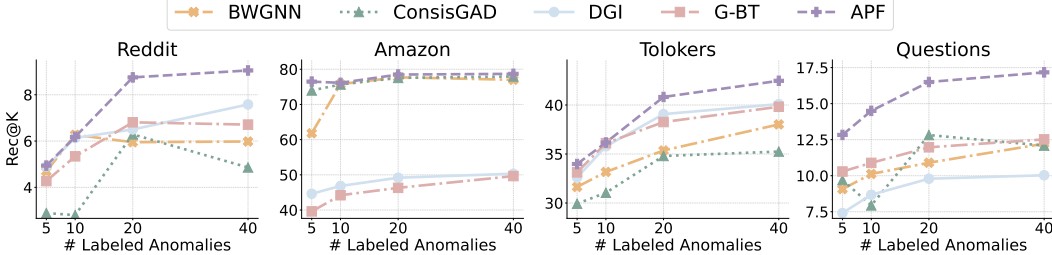

Figure 12: How the Rec@K score varies with different numbers of labeled anomalies.

## I.6 ADDITIONAL RESULTS FOR MODEL COMPARISON.

We provide additional results in terms of AUROC and Rec@K for model comparison in Table 7 and Table 8, respectively.

Table 7: Comparison of AUROC for each model. "-" denotes "out of memory". The best and runner-up models are **bolded** and underlined.

| Model | Reddit | Weibo | Amazon | Yelp. | T-Fin. | Ellip. | Tolo. | Quest. | DGraph. | T-Social | Avg. |
|---|---|---|---|---|---|---|---|---|---|---|---|
| GCN | 56.9±5.9 | 93.5±6.6 | 82.0±0.3 | 51.2±3.7 | 88.3±2.5 | 86.2±1.9 | 64.2±4.8 | 60.0±2.2 | 66.2±2.5 | 71.6±10.4 | 72.0 |
| GIN | 60.0±4.1 | 83.8±8.3 | 91.6±1.7 | 62.9±7.3 | 84.5±4.5 | 88.2±0.9 | 66.8±5.2 | 62.2±2.2 | 65.7±1.8 | 70.4±7.4 | 73.6 |
| GAT | 60.5±3.9 | 86.4±7.7 | 92.4±1.9 | 65.6±4.0 | 85.0±4.5 | 88.5±2.1 | 68.1±3.0 | 62.3±1.4 | 67.2±1.9 | 75.4±4.8 | 75.1 |
| ACM | 60.0±4.3 | 92.5±2.9 | 81.8±7.9 | 61.3±3.8 | 82.2±8.1 | 90.6±0.8 | 69.3±4.2 | 60.8±4.1 | 67.6±5.3 | 68.3±4.9 | 73.4 |
| FAGCN | 60.2±4.1 | 83.4±7.7 | 90.4±1.9 | 62.4±2.9 | 82.6±8.4 | 86.1±3.0 | 68.1±6.6 | 60.8±3.2 | 63.0±3.7 | - | - |
| AdaGNN | 62.0±4.7 | 69.5±4.8 | 90.8±2.2 | 63.2±3.1 | 83.6±2.6 | 85.1±2.8 | 63.3±5.3 | 58.5±4.1 | 67.6±3.7 | 64.7±5.6 | 70.8 |
| BernNet | 63.1±1.7 | 80.1±6.9 | 92.1±2.4 | 65.0±3.7 | 91.2±1.0 | 87.0±1.7 | 61.9±5.6 | 61.8±6.4 | 69.0±1.4 | 59.8±6.3 | 73.1 |
| GAS | 60.6±3.0 | 81.8±7.0 | 91.6±1.9 | 61.1±5.2 | 88.7±1.1 | 89.0±1.4 | 62.7±2.8 | 57.5±4.4 | 69.9±2.0 | 72.1±8.8 | 73.5 |
| DCI | 61.0±3.1 | 89.3±5.3 | 89.4±3.0 | 64.1±5.3 | 88.0±3.2 | 88.5±1.3 | 67.6±7.1 | 62.2±2.5 | 65.3±2.3 | 74.2±3.3 | 75.0 |
| PCGNN | 52.8±3.4 | 83.9±8.1 | 93.2±1.2 | 65.1±4.8 | 92.0±1.1 | 87.5±1.4 | 67.4±2.1 | 59.0±4.0 | 68.4±4.2 | 69.1±2.4 | 73.8 |
| AMNet | 62.9±1.8 | 82.4±4.6 | 92.8±2.1 | 64.8±5.2 | 92.6±0.9 | 85.4±1.7 | 61.7±4.1 | 63.6±2.8 | 67.1±3.2 | 53.7±3.4 | 72.7 |
| BWGNN | 57.7±5.0 | 93.6±4.0 | 91.8±2.3 | 64.3±3.4 | 92.1±2.7 | 88.7±1.3 | 68.5±2.7 | 60.2±8.6 | 65.5±3.1 | 77.5±4.3 | 76.0 |
| GHRN | 57.5±4.5 | 91.6±4.4 | 90.9±1.9 | 64.5±3.1 | 92.6±0.7 | 89.0±1.3 | 69.0±2.2 | 60.5±8.7 | 67.1±3.0 | 78.7±3.0 | 76.1 |
| ConsisGAD | 59.6±2.8 | 85.0±3.7 | 92.3±2.2 | 66.1±3.8 | 94.3±0.8 | 88.6±1.3 | 68.5±2.0 | 65.7±3.9 | 67.1±3.0 | 93.1±1.9 | 78.0 |
| SpaceGNN | 62.3±1.9 | 94.4±0.9 | 91.1±2.5 | 66.8±2.8 | 93.4±1.0 | 88.5±1.2 | 68.9±2.6 | 66.0±1.8 | 63.9±3.7 | 94.7±0.7 | 79.0 |
| XGBGraph | 59.2±2.7 | 96.4±0.7 | 94.7±0.9 | 64.0±3.5 | 94.8±0.6 | 91.9±1.3 | 67.5±3.4 | 61.4±2.9 | 62.4±4.1 | 85.2±1.8 | 77.8 |
| DGI | 63.0±3.0 | 96.7±2.5 | 87.7±0.7 | 54.0±1.8 | 91.8±0.8 | 86.2±1.4 | 71.5±0.7 | 68.7±3.8 | 64.3±2.2 | 89.3±1.3 | 77.3 |
| GRACE | 64.5±3.3 | 97.1±1.9 | 87.9±0.7 | 55.5±1.4 | 93.1±0.3 | 88.8±1.8 | 70.6±1.6 | 68.2±1.7 | - | - | - |
| G-BT | 63.8±4.3 | 97.3±1.1 | 84.8±1.6 | 55.5±1.8 | 93.0±0.6 | 88.5±1.2 | 71.7±1.3 | 67.0±1.6 | 69.4±2.5 | 90.9±1.2 | 78.2 |
| GraphMAE | 61.0±0.5 | 95.7±2.3 | 84.2±0.3 | 55.2±0.3 | 91.4±0.7 | 82.5±1.4 | 66.3±3.0 | 62.5±1.4 | 63.7±1.6 | 89.2±4.5 | 75.2 |
| BGRL | 65.3±2.2 | 99.0±0.5 | 84.3±1.0 | 57.0±1.2 | 88.0±1.7 | 88.5±1.7 | 72.0±1.8 | 65.7±2.9 | 64.9±1.6 | 90.2±1.3 | 77.5 |
| SSGE | 62.2±4.7 | 95.1±1.6 | 84.7±2.0 | 55.7±1.8 | 92.9±0.7 | 86.7±1.2 | 71.9±1.5 | 65.5±1.5 | 68.6±2.9 | 92.3±0.8 | 77.6 |
| PolyGCL | 62.7±3.9 | 97.4±0.6 | 93.3±1.3 | 65.1±4.1 | 89.3±0.6 | 87.9±0.9 | 68.1±1.8 | 64.8±3.8 | 67.4±2.0 | 91.4±1.5 | 78.7 |
| BWDGI | 60.0±3.5 | 91.4±0.8 | 94.1±1.6 | 66.9±2.7 | 94.0±0.6 | 87.6±1.5 | 71.4±1.9 | 64.0±3.9 | 69.8±1.8 | 82.8±2.6 | 78.2 |
| APF (*w/o* $\mathcal{L}_{pt}$) | 63.6±2.2 | 96.3±3.2 | 92.9±2.9 | 65.6±3.4 | 94.2±0.5 | 90.4±0.7 | 70.8±1.5 | 67.2±2.4 | 69.0±1.6 | 94.4±1.5 | 80.4 |
| **APF** | **66.8±3.9** | 98.8±0.3 | **94.9±1.1** | **68.2±2.3** | 94.8±0.5 | 91.2±0.9 | **73.7±1.0** | **71.9±2.1** | 72.4±1.3 | 95.1±1.4 | **82.8** |

Table 8: Comparison of Rec@K for each model. "-" denotes "out of memory". The best and runner-up models are **bolded** and underlined.

| Model | Reddit | Weibo | Amazon | Yelp. | T-Fin. | Ellip. | Tolo. | Quest. | DGraph. | T-Social | Avg. |
|---|---|---|---|---|---|---|---|---|---|---|---|
| GCN | 6.2±2.2 | 79.2±4.3 | 36.9±2.6 | 16.9±3.0 | 60.6±7.6 | 49.7±4.2 | 33.4±3.5 | 9.8±1.2 | 3.6±0.4 | 10.2±8.1 | 30.6 |
| GIN | 4.8±1.9 | 66.5±7.3 | 70.4±5.7 | 26.5±6.1 | 54.4±5.0 | 47.6±3.1 | 33.6±3.0 | 10.3±1.1 | 2.1±0.5 | 5.3±2.9 | 32.2 |
| GAT | 6.5±2.3 | 70.2±4.6 | 77.1±1.7 | 28.1±3.4 | 36.2±10.3 | 51.4±5.8 | 35.1±1.8 | 10.9±0.9 | 3.1±0.7 | 11.6±3.0 | 33.0 |
| ACM | 5.4±1.8 | 70.7±9.5 | 56.1±14.2 | 23.9±3.8 | 37.2±19.3 | 60.2±3.3 | 35.8±4.2 | 11.4±2.6 | 1.9±0.7 | 8.1±1.8 | 31.1 |
| FAGCN | 7.2±1.9 | 67.8±8.1 | 71.7±3.1 | 25.6±2.8 | 39.6±30.3 | 48.5±11.3 | 35.6±3.7 | 12.3±2.3 | 2.5±0.8 | - | - |
| AdaGNN | 6.3±2.2 | 38.3±3.7 | 74.2±4.0 | 25.6±2.4 | 31.3±11.3 | 46.3±7.4 | 33.6±3.7 | 10.0±2.4 | 1.1±0.4 | 7.9±2.7 | 27.5 |
| BernNet | 6.4±1.5 | 60.9±4.6 | 77.2±2.1 | 26.8±3.1 | 60.5±11.1 | 47.0±4.5 | 30.1±3.8 | 10.3±2.7 | 3.8±0.6 | 3.3±2.8 | 32.6 |
| GAS | 6.6±2.5 | 62.0±6.9 | 77.4±1.7 | 24.6±4.1 | 54.2±9.5 | 51.9±5.2 | 33.0±3.9 | 9.1±2.9 | 3.4±0.4 | 11.5±4.6 | 33.4 |
| DCI | 4.5±1.4 | 68.5±3.5 | 68.3±7.2 | 26.8±5.5 | 58.5±6.3 | 50.0±3.8 | 33.5±5.6 | 9.9±1.9 | 2.3±0.7 | 6.3±6.8 | 32.9 |
| PCGNN | 3.0±2.1 | 65.1±6.6 | 78.0±1.5 | 27.8±3.8 | 63.9±6.3 | 46.5±7.3 | 34.3±1.6 | 10.1±3.9 | 3.7±1.0 | 13.5±3.1 | 34.6 |
| AMNet | 6.8±1.5 | 62.1±4.4 | 77.8±2.3 | 26.6±4.3 | 65.7±6.3 | 37.8±6.7 | 30.5±1.9 | 12.7±2.6 | 2.6±0.8 | 1.6±0.5 | 32.4 |
| BWGNN | 6.0±1.4 | 75.1±3.5 | 77.7±1.6 | 26.4±3.2 | 64.9±11.7 | 49.7±6.1 | 35.5±3.1 | 10.9±3.2 | 3.1±0.8 | 24.3±7.4 | 37.4 |
| GHRN | 6.3±1.5 | 72.4±2.6 | 77.7±1.3 | 26.9±3.1 | 61.7±4.4 | 50.8±4.8 | 36.1±3.1 | 11.3±3.4 | 3.4±0.7 | 24.6±7.0 | 37.7 |
| ConsisGAD | 6.3±2.5 | 58.6±4.6 | 77.5±2.8 | 28.7±3.2 | 76.5±4.2 | 50.8±7.8 | 34.8±2.3 | 12.8±3.1 | 1.8±0.5 | 48.5±4.6 | 39.6 |
| SpaceGNN | 6.0±2.0 | 72.2±3.9 | 76.8±2.0 | 28.9±2.4 | 76.6±3.7 | 48.6±4.9 | 35.4±2.5 | 11.5±2.2 | 2.3±0.8 | 63.3±4.0 | 42.2 |
| XGBGraph | 4.9±1.9 | 68.9±5.7 | 78.2±1.5 | 26.8±3.0 | 72.4±3.8 | 68.9±3.7 | 36.6±3.0 | 10.6±2.9 | 2.5±0.7 | 43.0±7.6 | 41.3 |
| DGI | 6.5±1.3 | 85.5±2.1 | 49.2±2.2 | 18.8±1.2 | 71.7±4.8 | 48.0±2.1 | 39.1±1.1 | 9.8±2.8 | 3.1±0.7 | 43.2±4.3 | 37.5 |
| GRACE | 5.6±2.6 | 85.6±2.5 | 51.8±4.5 | 20.4±1.4 | 74.8±1.1 | 52.4±3.7 | 38.4±2.0 | 13.0±1.9 | - | - | - |
| G-BT | 6.8±1.7 | 85.1±3.6 | 46.3±3.4 | 20.6±1.5 | 74.0±1.9 | 50.7±3.7 | 38.3±2.8 | 12.0±3.4 | 3.8±0.6 | 46.2±6.0 | 38.4 |
| GraphMAE | 4.7±0.5 | 86.8±2.4 | 47.6±1.1 | 20.2±0.2 | 67.4±4.7 | 37.9±3.6 | 37.1±2.0 | 9.6±2.1 | 3.5±0.4 | 44.8±10.6 | 36.0 |
| BGRL | 7.4±1.1 | 90.3±1.1 | 45.3±3.7 | 21.6±1.5 | 59.3±3.4 | 51.0±4.7 | 38.2±3.1 | 11.5±3.6 | 2.7±0.9 | 47.1±5.2 | 37.4 |
| SSGE | 6.4±2.2 | 81.0±2.2 | 45.1±3.9 | 20.6±1.8 | 74.7±0.9 | 51.6±2.8 | 38.3±2.7 | 11.4±1.6 | 3.6±0.8 | 49.5±3.2 | 38.2 |
| PolyGCL | 7.4±2.5 | 81.7±2.9 | 72.5±7.5 | 27.5±2.9 | 50.4±3.8 | 55.0±2.5 | 35.6±1.8 | 9.0±2.4 | 1.9±0.7 | 44.7±5.1 | 38.6 |
| BWDGI | 6.2±1.7 | 64.5±1.7 | 72.2±7.7 | 29.8±2.9 | 75.6±2.7 | 50.3±5.4 | 39.5±2.5 | 7.5±1.5 | 3.1±0.6 | 43.3±3.8 | 39.2 |
| APF (*w/o* $\mathcal{L}_{pt}$) | 7.6±1.2 | 80.8±7.7 | 77.4±2.4 | 27.2±2.5 | 74.8±3.8 | 57.8±2.7 | 38.3±1.6 | 13.6±1.5 | 3.0±0.6 | 69.7±8.2 | 45.0 |
| **APF** | **8.8±1.6** | 88.7±2.4 | **78.5±4.2** | **31.4±1.6** | **78.9±1.3** | 62.1±2.1 | **40.8±1.7** | **16.5±0.9** | 4.2±0.7 | 74.1±5.4 | **48.4** |

### I.7 ADDITIONAL FIGURES FOR HOMOPHILY DISPARITY.

We provide additional figures in terms of AUROC and Rec@K for homophily disparity in Figure 13 and Figure 14, respectively.

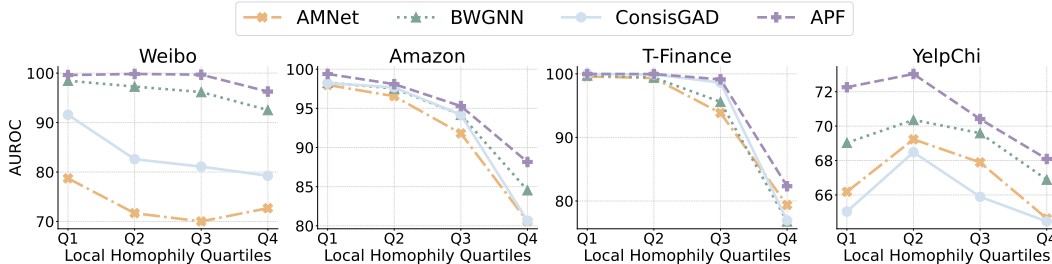

Figure 13: Performance across local homophily quartiles (Q1 = top 25%, Q4 = bottom 25%).

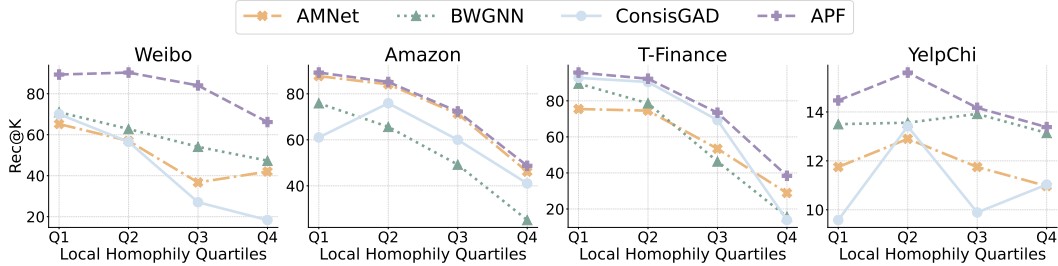

Figure 14: Performance across local homophily quartiles (Q1 = top 25%, Q4 = bottom 25%).

### I.8 PERFORMANCE ANALYSIS BY SUPERVISION PARADIGM

To clarify performance differences and better position our method within the landscape of GAD approaches, we provide an additional analysis categorizing baseline methods by the type of anomaly information used. While our main text categorizes models by architecture (e.g., Standard GNNs vs. Specialized GAD models), here we classify them based on their reliance on supervision, particularly under the label-scarce setting (100 labeled nodes) used in our experiments:

- **Supervised Models:** These models are trained directly using the available labeled anomalies. This category includes standard GNNs (*e.g.*, GCN, GAT, ACM) and supervised GAD-specific models (*e.g.*, GAS, PCGNN, AMNet, BWGNN, GHRN).

- **Semi-supervised Models:** These are specialized subclasses of supervised methods designed to effectively leverage limited anomaly labels. This category includes ConsisGAD, SpaceGNN, and XGBGraph.

- **Self-supervised Models:** These models adopt a two-stage paradigm: first pre-training on unlabeled data to learn general representations, followed by fine-tuning with labeled anomalies. This category includes general graph pre-training methods (*e.g.*, DGI, DCI, GraphMAE, SSGE, PolyGCL, BWDGI) and our proposed **APF**.

Table 9 summarizes the performance of the top-3 models from each category alongside APF. This categorization yields two critical observations regarding label scarcity in GAD:

1. **Potential of Pre-training:** In this realistic label-scarce scenario, general-purpose self-supervised models (e.g., BWDGI, SSGE) are competitive with, and often outperform, the best specialized supervised GAD models (e.g., GHRN, BWGNN). For instance, BWDGI achieves an average AUPRC of 39.2% compared to GHRN's 35.4%. This confirms the advantage of the pre-training paradigm in extracting transferable knowledge from abundant unlabeled data when supervision is limited.

Table 9: Comparison of top-performing models across different supervision paradigms. Results are averaged across all 10 datasets. Our proposed APF (Self-supervised) demonstrates superior performance compared to the best models in Supervised and Semi-supervised categories.

| Category | Model | Avg. AUPRC | Avg. AUROC | Avg. Rec@K |
|---|---|---|---|---|
| **Supervised** | PCGNN | 32.9 | 73.8 | 34.6 |
| | BWGNN | 35.4 | 76.0 | 37.4 |
| | GHRN | 35.4 | 76.1 | 37.7 |
| **Semi-supervised** | ConsisGAD | 38.6 | 78.0 | 39.6 |
| | SpaceGNN | 41.8 | 79.0 | 42.2 |
| | XGBGraph | 42.9 | 77.8 | 41.3 |
| **Self-supervised** | SSGE | 37.1 | 77.6 | 38.2 |
| | PolyGCL | 37.1 | 78.7 | 38.6 |
| | BWDGI | 39.2 | 78.2 | 39.2 |
| | **APF (Ours)** | **49.6** | **82.8** | **48.4** |

2. **Effectiveness of Anomaly-Aware Design:** While general self-supervised models show promise, our proposed APF significantly outperforms them, as well as the strongest semi-supervised baselines. APF achieves an average AUPRC of 49.6%, surpassing the best semi-supervised model (XGBGraph, 42.9%) by 6.7% and the best baseline self-supervised model (BWDGI, 39.2%) by 10.4%. This validates our core motivation: while the pre-training paradigm is beneficial, a general-purpose objective is insufficient. A framework specifically tailored to capture anomaly-aware signals, as APF does via the Rayleigh Quotient and dual-filter encoding, is essential for maximizing performance in GAD.

## I.9 COMPARISON WITH JOINT LEARNING STRATEGY

To justify our design choice of a two-stage framework (pre-training followed by fine-tuning), we compare our proposed method against a *joint learning* strategy. We define the two strategies as follows:

- **Two-stage (Ours):** The model is first trained with the unsupervised pre-training objective. The resulting representations are then frozen or used as initialization for fine-tuning with the supervised binary classification loss.

- **Joint Learning:** The model is trained end-to-end by simultaneously optimizing both the supervised classification loss and the unsupervised pre-training loss (i.e., $\mathcal{L}_{total} = \mathcal{L}_{sup} + \lambda\mathcal{L}_{unsup}$).

We applied these strategies to standard baselines (GCN and BWGNN, using DGI as the auxiliary unsupervised objective) as well as to our APF framework. Table 10 presents the AUPRC performance across four representative datasets.

Table 10: Performance comparison (AUPRC %) between Supervised-only, Joint Learning, and Two-stage strategies. The two-stage paradigm consistently outperforms joint learning, with APF achieving the best overall results.

| Model | Training Strategy | Reddit | YelpChi | Tolokers | DGraph-Fin |
|---|---|---|---|---|---|
| GCN | Supervised-only | $4.2 \pm 0.8$ | $16.4 \pm 2.6$ | $33.0 \pm 3.6$ | $2.3 \pm 0.2$ |
| | Joint Learning (w/ DGI) | $4.4 \pm 1.0$ | $18.2 \pm 2.3$ | $38.0 \pm 3.4$ | $2.3 \pm 1.4$ |
| | Two-stage (w/ DGI) | $4.8 \pm 0.6$ | $17.0 \pm 1.2$ | $39.7 \pm 0.8$ | $2.1 \pm 0.2$ |
| BWGNN | Supervised-only | $4.2 \pm 0.7$ | $23.7 \pm 2.9$ | $35.3 \pm 2.2$ | $2.1 \pm 0.3$ |
| | Joint Learning (w/ DGI) | $4.3 \pm 0.6$ | $26.8 \pm 4.1$ | $38.2 \pm 3.4$ | $2.3 \pm 0.2$ |
| | Two-stage (w/ DGI) | $4.5 \pm 0.6$ | $26.8 \pm 2.7$ | $38.5 \pm 3.1$ | $2.4 \pm 0.2$ |
| APF | Supervised-only (w/o $\mathcal{L}_{pt}$) | $5.2 \pm 0.6$ | $24.1 \pm 2.2$ | $37.4 \pm 1.2$ | $2.3 \pm 0.2$ |
| | Joint Learning | $5.5 \pm 0.7$ | $28.0 \pm 1.8$ | $39.6 \pm 2.2$ | $2.7 \pm 0.3$ |
| | **Two-stage (Ours)** | $\mathbf{5.9 \pm 0.9}$ | $\mathbf{28.4 \pm 1.4}$ | $\mathbf{40.5 \pm 2.0}$ | $\mathbf{2.9 \pm 0.2}$ |

The results yield two critical insights:

1. **Benefits of Unsupervised Signals:** Consistent with our hypothesis, incorporating unsupervised objectives (whether via joint learning or two-stage training) generally improves performance over purely supervised baselines. For example, GCN with Joint Learning improves upon the supervised GCN on Reddit and YelpChi.

2. **Superiority of Two-Stage Learning:** In nearly all cases, the two-stage paradigm outperforms the joint learning strategy. This trend is particularly pronounced in APF, where the two-stage approach achieves the highest performance across all datasets. We attribute this to the potential conflict between the unsupervised pre-training objective and the supervised classification loss when optimized simultaneously, which may lead to suboptimal representations. This observation aligns with prior findings in GAD literature (Wang et al., 2021), which suggest that decoupling representation learning from classification often yields superior detection performance.

## J    DISCUSSION ON THEORETICAL ASSUMPTIONS AND APPLICABILITY

In Section 3.3, we establish the linear separability of anomalies under node-adaptive filtering using the Anomalous Stochastic Block Model (ASBM). We acknowledge that this theoretical model relies on certain idealizations compared to the deployed APF architecture. Here, we clarify the scope of these assumptions and their connection to the practical implementation.

**Gaussian Feature Assumption.** Our theoretical analysis assumes Gaussian-distributed node features to ensure analytical tractability and derive closed-form separability conditions. This assumption is standard in theoretical analyses of GNNs and GAD to isolate the effects of structural properties like homophily (Baranwal et al., 2021; Ma et al., 2022; Mao et al., 2023; Han et al., 2024). While real-world datasets such as YelpChi and T-Finance contain heterogeneous or categorical features (Tang et al., 2022), the ASBM serves as a simplified "sandbox" to demonstrate the efficacy of node-adaptive low-/high-pass filtering under homophily disparity. Our empirical results on these non-Gaussian datasets (Table 1) suggest that the architectural insights derived from this Gaussian setting are robust and transferable to more complex, real-world distributions.

**Oracle Patterns vs. Data-Driven Fusion.** Theorem 1 assumes an idealized scenario where node homophily patterns are known, allowing for the precise assignment of low-pass or high-pass filters. In practice, APF replaces this oracle assignment with the Gated Fusion Network (GFN) and anomaly-aware regularization. Specifically, the GFN generates continuous coefficients $C$ to create a soft, learnable relaxation of the hard filter assignment used in the theorem. The regularization loss $\mathcal{L}_{reg}$ further encourages the model to mimic the theoretical ideal by guiding abnormal nodes to rely more on the anomaly-sensitive (high-pass) branch. Visualizations of the learned coefficients (Figure 4) confirm that APF successfully approximates this ideal allocation in a data-driven manner.

**Linear vs. Deep Architectures.** Finally, while the theorem proves the existence of a linear separator on frozen filtered features, APF employs learnable polynomial filters and MLP encoders. The theoretical result is intended to provide a conceptual justification for the core mechanism of APF: the node-specific combination of low-pass and high-pass information. By proving that a linear classifier suffices under ideal filtering, we motivate the design of APF, which employs a more expressive parameterized implementation to learn these optimal filters and fusion strategies.

## K    ANALYSIS OF RAYLEIGH QUOTIENT ON DIFFERENT ANOMALY TYPES

To further justify our use of the Rayleigh Quotient (RQ) as a label-free anomaly indicator, we analyze its sensitivity to different graph anomaly types. The "right-shift" phenomenon, where spectral energy concentrates on high frequencies, is a fundamental indicator of anomaly degree (Tang et al., 2022). Here, we clarify how this phenomenon captures both *attribute anomalies* and *structural anomalies* through the lens of graph signal smoothness.

The Rayleigh Quotient is defined as $RQ(\boldsymbol{x}, \boldsymbol{L}) = \frac{\boldsymbol{x}^\top \boldsymbol{L} \boldsymbol{x}}{\boldsymbol{x}^\top \boldsymbol{x}}$ (Tang et al., 2022). The numerator, $\boldsymbol{x}^\top \boldsymbol{L} \boldsymbol{x} = \sum_{i,j} A_{ij}(x_i - x_j)^2$, quantifies the "smoothness" or consistency of the node attributes $\boldsymbol{x}$

with respect to the graph structure $\boldsymbol{L}$. A higher RQ value indicates a "right-shift" in spectral energy, signifying a high level of inconsistency. Both primary anomaly types contribute to this inconsistency:

- **Abnormal Node Attributes:** In this scenario, a node $v_i$ possesses feature values $x_i$ that deviate significantly from the distribution of its neighbors $x_j$. This creates a large feature difference $(x_i - x_j)^2$ across edges connected to $v_i$. Consequently, the term $\boldsymbol{x}^\top \boldsymbol{L} \boldsymbol{x}$ increases, resulting in a higher Rayleigh Quotient and a shift toward high-frequency spectral energy. This aligns with the theoretical analysis of Gaussian anomalies provided by Tang et al. (2022).
- **Abnormal Edge Connections (Structural Anomalies):** This scenario typically involves "camouflaged" anomalies, where an abnormal node intentionally connects to benign (normal) nodes to evade detection (Tang et al., 2022). While the node's features might appear valid in isolation, the *connection* creates an edge between dissimilar classes (anomalous vs. normal). Because the features of the anomalous node are inherently different from those of the normal community it has invaded, the term $(x_i - x_j)^2$ along these spurious edges becomes large. This breakage of homophily similarly increases the $\boldsymbol{x}^\top \boldsymbol{L} \boldsymbol{x}$ term, manifesting as a "right-shift" in the spectrum.

Both attribute and structural anomalies fundamentally break the smoothness assumption of the graph signal, leading to a higher concentration of spectral energy in the high-frequency domain. This universality makes the Rayleigh Quotient a robust, unified, and label-free metric for our pre-training stage, allowing APF to effectively target node-specific subgraphs that exhibit feature-structure mismatches regardless of the anomaly's origin.

## L  DETAILS OF DUAL-FILTER ENCODING

In this section, we elaborate on the design rationale and implementation details of the Dual-filter Encoding module introduced in Section 3.1.

**Motivation.** The core motivation behind our dual-filter design stems from the inherent complexity of the GAD task, which requires the simultaneous extraction of two distinct types of information:

- **General Semantic Patterns:** These represent the "normality" of the graph, where connected nodes typically share similar features (homophily). Such patterns are concentrated in the low-frequency range of the graph spectrum and are well-modeled by conventional low-pass filters used in standard GNNs.
- **Subtle Anomaly Cues:** Anomalies often manifest as high-frequency signals, characterized by abrupt changes in features across edges (heterophily) or structural inconsistencies. Relying solely on low-pass filters tends to smooth out these critical high-frequency signals, making anomalies indistinguishable from normal nodes.

To address this, APF complements the low-pass encoder with an explicit high-pass encoder. This ensures that while $\boldsymbol{Z}_L$ captures the general semantic structure, $\boldsymbol{Z}_H$ preserves the subtle anomaly cues, providing a comprehensive basis for the subsequent fusion module.

**Spectral Guarantees.** To implement these filters efficiently while retaining flexibility, we utilize the learnable Chebyshev polynomial approximation restricted by specific constraints on the coefficients. As defined in Eq. 7, the filter values are derived from a shared parameter vector $\boldsymbol{\gamma}$ via cumulative summation: $\gamma_k^L = \gamma_0 - \sum_{j=1}^{k} \gamma_j$ and $\gamma_k^H = \sum_{j=0}^{k} \gamma_j$.

This construction provides a theoretical guarantee on the spectral behavior of the filters. As demonstrated in prior work (Chen et al., 2024a), this formulation enforces the monotonicity of the filter response values at the Chebyshev nodes:

$$\gamma_i^L \geq \gamma_{i+1}^L, \quad \gamma_i^H \leq \gamma_{i+1}^H. \tag{17}$$

These inequalities ensure that $g_L(\cdot)$ consistently attenuates high frequencies (low-pass property) while $g_H(\cdot)$ amplifies them (high-pass property), preventing the optimization process from collapsing into arbitrary or redundant filter shapes.

