# OpenReview forum: "Towards Anomaly-Aware Pre-Training and Fine-Tuning for Graph Anomaly Detection"
_ICLR.cc/2026/Conference — ICLR 2026 Poster_

### Official Review · Reviewer_TPKe · 2025-10-18

**Soundness:** 3
**Presentation:** 3
**Contribution:** 2
**Rating:** 6
**Confidence:** 3

**Summary:**

The authors introduce Anomaly-Aware Pre-Training and Fine-Tuning (APF), a two stage framework for supervised GAD. Experiments show the effectiveness of the framework.

**Strengths:**

1. The authors introduce Anomaly-Aware Pre-Training and Fine-Tuning (APF), a two-stage framework for supervised GAD.
2. Experiments show the effectiveness of the framework.

**Weaknesses:**

1. The novelty can be a question, as all the components are from existing work, which means such a framework can be considered an implementation rather than a contribution.
2. How to choose the hyperparameters for new datasets can be a question since the combinations of hyperparameters require extensive experiments for grid search, as shown in Appendix H.4.

**Questions:**

Please refer to the weaknesses.

---

> ### Author Response · Authors · 2025-11-19
> **Official Response by Authors (1)**
>
> Thanks for your valuable suggestions. We will try to address your concerns, and we are eager to engage in a more detailed discussion with you.
>
> > W1: Limited Perceived Novelty
>
> Thank you for this critical question. It is an important point of clarification, and we appreciate the opportunity to clarify our core contribution. The individual components we use, such as the Rayleigh Quotient (RQ) and spectral polynomial filters, are well-established. Our contribution, however, is **not in the invention of these components, but in the new, principled framework we have designed *by* adapting and co-designing them to solve a critical research problem.**
>
> Our work is motivated by a key finding: general-purpose, *task-agnostic* pre-training can already match or beat many *GAD-specific* supervised models in label-scarce settings. This pointed to a significant, untapped potential: **what if we designed a pre-training and fine-tuning paradigm that was *explicitly* GAD-specific and anomaly-aware from the start?**
>
> This is the core problem we address, which, to our knowledge, has not been well-studied. Our framework is not a trivial "implementation" but a *principled solution* to this new problem, where each piece is a necessary and non-obvious adaptation:
>
> **1. An Anomaly-Aware Pre-Training Objective**
>
> This is our first key technical contribution. We identified that existing pre-training paradigms lack an objective that is "anomaly-aware."
>
> - **Prior Art:** The RQ has been used in *supervised* GAD models [1, 2].
>
> - **Our Contribution:** We are the first to adapt the RQ as a **label-free anomaly indicator** to guide an **unsupervised pre-training objective**. This objective, which maximizes mutual information between a high-pass filter and RQ-guided subgraphs, is a new pre-training task formulation designed to capture subtle anomaly cues *before any labels are seen*.
>
> **2. A Necessary and Co-designed Architecture**
>
> This new objective *requires* a specific backbone. A standard low-pass GNN (used in general pre-training) cannot be effectively trained by our anomaly-aware objective.
>
> - **Our Contribution:** The **Dual-filter Encoding**  is the necessary architectural counterpart to our dual-objective. It is designed to learn and, critically, **decouple** the representations for general semantics ($Z_L$) and anomaly-specific cues ($Z_H$). This decoupling is a core part of our framework's design.
>
> **3. A Synergistic Fine-Tuning Stage**
>
> This two-stage design is not a simple combination; the fine-tuning stage is specifically engineered to *exploit* the decoupled representations learned in stage one.
>
> - **Our Contribution:** The **Gated Fusion Network** and **Anomaly-Aware Regularization Loss** are designed to *adaptively integrate* the pre-trained $Z_L$ and $Z_H$ representations. This mechanism is our solution for handling the homophily disparity challenge, allowing the model to decide how much "semantic" vs. "anomaly" information to trust for each node.
>
> **4. An Theoretical Justification**
>
> Finally, this entire framework is not ad-hoc; it is **theoretically motivated**.
>
> - **Our Contribution:** We introduce a new graph generative model, the **Anomalous Stochastic Block Model (ASBM)**, which is a variant of CSBM specifically designed to reflect GAD challenges like homophily disparity and class imbalance .
> - Under this model, our **Theorem 1** proves that linear separability is achievable if, and only if, low-pass and high-pass filters are *adaptively applied* to different nodes based on their local homophilic or heterophilic patterns.
> - This theory provides the formal justification for our architecture: the dual-filter encoding *provides the filter options*, and the gated fusion network *learns the adaptive assignment*, creating a practical framework that approximates this theoretical path to linear separability.
>
> In summary, the introduction of these techniques is not a trivial combination, but a **co-designed, synergistic, and theoretically-supported solution** to a new research problem. The technical contribution is the design of this *entire anomaly-aware paradigm*, which is validated by its superior performance. Thank you again for this insightful comment.

---

> ### Author Response · Authors · 2025-11-19
> **Official Response by Authors (2)**
>
> > W2: Hyperparameter Selection Concerns
>
> Thank you for raising this important practical concern.
>
> You are correct that the hyperparameter grids shown in Appendix were intentionally extensive. We included this full grid search for the sake of thoroughness and to provide a complete analysis of the model's behavior across the entire parameter space, respecting the constraints $p_n, p_a \in [0, 1]$ and $p_a \le p_n$.
>
> However, this exhaustive search is **not required in practical use**. Our own analysis in Appendix (Figures 7-9) reveals a very strong and stable pattern: optimal performance is consistently achieved when $p_n$ (the target preference for *normal* nodes) is set to a high value (e.g., $\ge 0.9$). This aligns with our model's intuition that normal nodes should rely heavily on the generic, low-pass semantic representations.
>
> In practice, hyperparameter tuning is much simpler. As reflected in the `train.conf.yaml` configuration file included in our anonymous code, we recommend a highly efficient strategy for new datasets:
>
> 1. Fix $p_n$ to a high default value (e.g., 0.9 or 1.0).
> 2. Perform a simple search for $p_a$ over a very small set, such as {0.1, 0.2, 0.3, 0.4}, ensuring $p_a \le p_n$.
>
> This reduces the effective hyperparameter search space from a large grid to just a few values, making our method practical and easy to adapt to new datasets.
>
> **ACTION**: We have clarified this practical tuning process in Appendix H.4 Implementation Details. Thank you again for this constructive feedback.

---

> > ### Comment · Reviewer_TPKe · 2025-11-27
> >
> > Thanks for the rebuttal. Although I still have concerns about the novelty and practical use, I understand opinions can vary, so I have decided to keep my score.

---

> > > ### Author Response · Authors · 2025-11-27
> > > **Thank you again!**
> > >
> > > Thank you very much for taking the time to re-evaluate our rebuttal and for sharing your updated thoughts.
> > >
> > > We sincerely appreciate your willingness to consider our clarifications, and we fully understand that perspectives on novelty and practical usability can naturally vary across researchers. Your comments have helped us better articulate our motivation, the technical design choices, and the practical implications of our work, and we are grateful for the opportunity to refine our presentation accordingly.
> > >
> > > Regardless of the final outcome, your feedback has been genuinely valuable for improving the paper. We have already incorporated the clarifications on (1) the conceptual contribution of an anomaly-aware pre-training paradigm and its theoretical grounding, and (2) the simplified hyperparameter tuning procedure for real-world use. These improvements will, we believe, make the paper more accessible and practically relevant for the community.
> > >
> > > Thank you again for your thoughtful review, constructive suggestions, and professional attitude throughout the discussion. We truly appreciate it.

---

### Official Review · Reviewer_EFtV · 2025-11-01

**Soundness:** 3
**Presentation:** 3
**Contribution:** 3
**Rating:** 6
**Confidence:** 4

**Summary:**

This paper addresses the challenges of label scarcity and homophily disparity in graph anomaly detection (GAD). It proposes a novel framework called Anomaly-Aware Pre-Training and Fine-Tuning (APF), which incorporates a label-free anomaly metric (Rayleigh Quotient) and dual spectral filters during pre-training to capture both semantic and anomaly-sensitive signals. For fine-tuning, APF employs a gated fusion mechanism and an anomaly-aware regularization loss to adaptively handle node- and class-level homophily disparities.

**Strengths:**

1. Drawing on recent research findings, this paper provides an in-depth analysis of local data issues in graph anomaly detection.

2.It creatively applies Rayleigh Quotient to node-level subgraph selection and demonstrates its effectiveness through experiments.

3.Experiments span 10 diverse datasets, ensuring robustness and generalizability across domains such as social networks and finance.

**Weaknesses:**

1. The authors only cite the relationship between spectral energy (i.e., the quantification of the "right shift" phenomenon) and anomaly degree, but do not clearly discuss its connection to graph anomaly detection. For example, is this anomaly due to abnormal node attributes or abnormal edge connections? Will the two types of anomalies differ in spectral density / energy?

2. Considering the use of the MRQSampler algorithm in this paper, the authors should add a discussion of UniGAD to clarify the paper's distinction and contribution.

3. The design of the Dual-filter Encoding lacks detailed explanation.

**Questions:**

Please refer to the weaknesses.

---

> ### Author Response · Authors · 2025-11-19
> **Official Response by Authors (1)**
>
> Thanks for your valuable suggestions. We will try to address your concerns, and we are eager to engage in a more detailed discussion with you.
>
> > W1: Insufficient Explanation of How Spectral Energy Relates to Different Anomaly Types
>
> Thank you for this insightful question. It touches on the core of our motivation for using the Rayleigh Quotient as a label-free anomaly indicator. The "right shift" phenomenon, which we quantify using the Rayleigh Quotient (RQ), is not limited to a single type of anomaly (e.g., attribute vs. structure). Instead, it provides a fundamental measure of the **inconsistency between node attributes and the local graph structure**.
>
> The RQ is defined as $RQ(x, L) = \frac{x^T L x}{x^T x}$ [1]. The numerator, $x^T L x$, quantifies the "smoothness" of the signal $x$ over the graph $L$. A low value means connected nodes have similar features, while a high value (a "right shift") means connected nodes are dissimilar.
>
> To answer your specific questions:
>
> 1. **Is this anomaly due to abnormal attributes or abnormal edge connections?** It captures both, as they both manifest as a feature-structure inconsistency.
>    - **Abnormal Node Attributes:** This is the case analyzed theoretically in the BWGNN paper [1]. If a node has highly abnormal features ($x_i$), it will be very different from its normal neighbors ($x_j$). This creates a large difference $(x_i - x_j)^2$ for each of its connections, which increases the $x^T L x$ term and results in a "right shift".
>    - **Abnormal Edge Connections (Structural Anomalies):** This scenario is also captured. Consider an anomalous node that "camouflages" itself by *intentionally connecting to many benign (normal) nodes* [1]. These "abnormal" edges create connections between nodes that are, by definition, dissimilar in their true nature (anomalous vs. normal). This dissimilarity again increases the $x^T L x$ term, causing a "right shift".
> 2. **Will the two types of anomalies differ in spectral density / energy?** Both types of anomalies will contribute to a **higher concentration of spectral energy in the high-frequency part of the spectrum** (i.e., a "right shift") [1].
>    - An anomaly defined by its *attributes* breaks the smoothness assumption, pushing energy to high frequencies.
>    - An anomaly defined by its *connections* (e.g., connecting to a "normal" community it doesn't belong to) also breaks the smoothness assumption.
>
> This is precisely **why we selected the Rayleigh Quotient-guided subgraph sampling for our pre-training stage**. It serves as a unified, label-free metric that captures the fundamental *mismatch* between features and structure, which is the hallmark of most graph anomalies, regardless of whether they originate from abnormal attributes or camouflaged connections.The existing theoretical and empirical evidence in [1] provides a solid foundation for our design choice, allowing us to target these anomalies within a unified pre-training objective.
>
> **ACTION**: We have included this analysis to Appendix K Analysis of Rayleigh Quotient on Different Anomaly Types in the revised manuscript. Thank you again for your valuable feedback.
>
> [1] Tang et al. *Rethinking Graph Neural Networks for Anomaly Detection.* ICML 2022

---

> ### Author Response · Authors · 2025-11-19
> **Official Response by Authors (2)**
>
> > W2: Missing Discussion on Relation to UniGAD
>
> Thank you for this valuable suggestion. UniGAD is indeed a highly relevant and concurrent work, and we appreciate the opportunity to clarify the important distinctions between our APF framework and their approach.
>
> The two frameworks, while both leveraging the Rayleigh Quotient, are designed to solve fundamentally different problems with distinct technical contributions.
>
> 1.  **Different Core Objectives:**
>     * UniGAD's primary contribution is a *unified framework* designed to jointly detect anomalies at the **node, edge, and graph levels**. It addresses the challenge of *multi-level GAD*, where existing methods typically focus on only a single object type (e.g., just nodes).
>     * Our APF's contribution is a *pre-training and fine-tuning framework* specifically designed to tackle **label scarcity** and **homophily disparity** in *single-level node anomaly detection*.
>
> 2.  **Different Roles for the MRQSampler:** The most important distinction lies in *how* each paper uses the Maximum Rayleigh Quotient Sampler (MRQSampler).
>     * In UniGAD, the MRQSampler is a *core architectural component* whose purpose is to **unify task formats**. It transforms node-level and edge-level tasks into graph-level tasks by sampling subgraphs. This allows all three task types (node, edge, graph) to be processed by a single, unified model.
>     * In our APF, we use the MRQSampler as a *tool* within our **unsupervised pre-training stage**. Its purpose is not to unify tasks, but to *inject anomaly-awareness* into our high-pass spectral filter *before* fine-tuning begins. The sampler is used to construct an anomaly-sensitive objective that complements the standard semantic-focused (low-pass) objective.
>
> 3.  **Different Core Technical Novelties:**
>     * The core novelty of UniGAD lies in its **unification mechanisms**: the MRQSampler (for unifying formats) and the GraphStitch Network (for unifying multi-level training).
>     * The core novelty of our APF lies in its **anomaly-aware two-stage design**. This includes: (1) An anomaly-aware pre-training objective, which is distinct from prior task-agnostic methods. It explicitly injects anomaly-awareness by maximizing mutual information between a high-pass filter and the MRQ-guided subgraphs. (2) A granularity-adaptive fine-tuning framework that uses a gated fusion network and an anomaly-aware regularization loss to adaptively combine the pre-trained semantic ($Z_L$) and anomaly-specific ($Z_H$) representations.
>
> In short, UniGAD uses the RQ sampler to *build* a unified multi-level model. We use the RQ sampler to *pre-train* anomaly-aware representations that are then fed into our novel granularity-adaptive fine-tuning framework to solve label scarcity.
>
> **ACTION**: We have added this discussion in Appendix C Related Works to clearly situate our contribution. Thanky you again for pointing out this important connection.

---

> ### Author Response · Authors · 2025-11-19
> **Official Response by Authors (3)**
>
> > W3: Insufficient Explanation of the Dual-filter Encoding Design
>
> Thank you for the constructive suggestion. We will expand the description of our Dual-filter Encoding design in the revised manuscript. Here is a more detailed explanation:
>
> **Motivation for Dual-filter Encoding:** Our design is motivated by the fact that graph anomaly detection (GAD) requires capturing two types of information: (1) **general semantic patterns** (i.e., "normality"), which are well-modeled by conventional low-pass filters, and (2) **subtle anomaly cues**, which often manifest as high-frequency signals or heterophilic patterns. Relying on only a low-pass filter, as in standard GNNs, can smooth out these critical anomaly signals. Therefore, we complement the low-pass encoder with an additional high-pass encoder to better capture these cues.
>
> **Filter Implementation via Chebyshev Polynomials:** To implement these encoders with flexible spectral properties, we adopt the learnable K-order Chebyshev polynomial approximation. Following prior work4, we restrict this flexible polynomial to fit only a low-pass filter, $g_{L}(\cdot)$, and a high-pass filter, $g_{H}(\cdot)$.
>
> The filters are formulated as:
> $$
> g_{L}(\hat{L})=\sum_{k=0}^{K}w_{k}^{L}T_{k}(\hat{L}), \quad g_{H}(\hat{L})=\sum_{k=0}^{K}w_{k}^{H}T_{k}(\hat{L}),
> $$
> where:
>
> - $\hat{L}=2\tilde{L}/\lambda_{n}-I$ is the scaled Laplacian matrix.
> - $T_{k}(x)$ are the Chebyshev polynomials, recursively defined as $T_{k}(x)=2xT_{k-1}(x)-T_{k-2}(x)$ with $T_{0}(x)=1$ and $T_{1}(x)=x$.
>
> **Learnable Coefficients and Spectral Guarantee:** The key to specializing these filters lies in how their coefficients ($w_k^L, w_k^H$) are computed from a shared learnable parameter vector $\gamma = (\gamma_0, \dots, \gamma_M)$. The coefficients are first determined by filter values $\gamma_k^L$ and $\gamma_k^H$ at specific Chebyshev nodes.
>
> These filter values are defined as:
> $$
> \gamma_{k}^{L}=\gamma_{0}-\sum_{j=1}^{k}\gamma_{j} \quad \text{and} \quad \gamma_{k}^{H}=\sum_{j=0}^{k}\gamma_{j},
> $$
> where $\gamma_0$ is shared. As demonstrated by Chen et al. (2024a), this formulation ensures that $\gamma_{i}^{L} \ge \gamma_{i+1}^{L}$ (guaranteeing the low-pass property for $g_L$) and $\gamma_{i}^{H} \le \gamma_{i+1}^{H}$ (guaranteeing the high-pass property for $g_H$).
>
> **Final Encoder Formulation:** This design allows our encoders to learn the optimal low-pass and high-pass filters for the data during pre-training. The final low-pass ($Z_L$) and high-pass ($Z_H$) node representations are then generated by passing the filtered features through respective MLPs ($f_{\theta_L}, f_{\theta_H}$):
> $$
> Z_{L}=f_{\theta_{L}}(g_{L}(\hat{L})X), \quad Z_{H}=f_{\theta_{H}}(g_{H}(\hat{L})X),
> $$
> These dual representations, $Z_L$ (capturing general semantics) and $Z_H$ (capturing anomaly cues), form the foundation for our anomaly-aware pre-training and subsequent granularity-adaptive fine-tuning.
>
> **ACTION**: We have provided more details in Section 3.1 Dual-filter Encoding and Appendix L Details of Dual-filter Encoding. Thank you again for this constructive feedback.

---

> > ### Comment · Reviewer_EFtV · 2025-11-27
> >
> > Thanks for the replies. These discussions have made this paper much clearer. After reading the comments from other reviewers, I decided to keep my score.

---

> > > ### Author Response · Authors · 2025-11-27
> > > **Thank you again!**
> > >
> > > Thank you again for your constructive review and support. It truly helps us improve the clarity and rigor of our work.

---

### Official Review · Reviewer_sLnc · 2025-11-01

**Soundness:** 2
**Presentation:** 2
**Contribution:** 2
**Rating:** 2
**Confidence:** 4

**Summary:**

This paper addresses graph anomaly detection under label scarcity by introducing APF, a two-stage framework that combines anomaly-aware pre-training with granularity-adaptive fine-tuning. The core methodology consists of: (i) a pre-training stage that uses the Rayleigh Quotient to guide subgraph sampling and employs both low-pass and high-pass learnable filters to capture both semantic and anomaly-specific signals, and (ii) a fine-tuning stage that adaptively fuses these representations via a gated fusion network with anomaly-aware regularization. The authors provide theoretical analysis under an Anomalous Stochastic Block Model to demonstrate potential linear separability and conduct extensive experiments on 10 benchmark datasets to validate the effectiveness of APF.

**Strengths:**

**S1**. The paper articulates the challenges in GAD, particularly label scarcity and homophily disparity, in a clear and accessible manner. The motivation for using the Rayleigh Quotient as a label-free anomaly indicator is well-explained, and the distinction between local and global homophily effectively highlights limitations of uniform processing schemes. The paper is easy to follow.

**S2**. The incorporation of the Rayleigh Quotient for subgraph selection and the use of separate high-pass and low-pass spectral filters represent sensible design choices for capturing both general semantic patterns and anomaly-specific signals.

**S3**. The authors conduct extensive experiments across 10 diverse real-world datasets.

**Weaknesses:**

**W1**. While the paper combines several techniques, the individual components are largely incremental adaptations of well-established methods. The Rayleigh Quotient has been extensively used in prior GAD work [1,2,3], and spectral filtering with Chebyshev polynomials is standard practice [4], representing a modest modification rather than a fundamental innovation. The paper would benefit from more clearly articulating what specific technical innovations go beyond combining existing techniques and why this particular combination is uniquely effective.

**W2**. The theoretical analysis in Section 3.3 and Theorem 1 suffers from several critical limitations that undermine its relevance to the actual method: (i) The ASBM assumes Gaussian-distributed node features, which is unrealistic for most real-world datasets that contain heterogeneous, sparse, or even categorical features; (ii) The theoretical setup assumes perfect knowledge of node homophily patterns and proposes applying filters accordingly, whereas the actual method uses a data-driven gated fusion network that does not explicitly model or utilize such discrete pattern assignments; (iii) The proof relies on a linear classifier with frozen filtered features, but APF uses learnable polynomial filters, MLP encoders, and a trainable fusion mechanism, creating multiple layers of learnable transformations that are not captured by the theoretical model. These gaps make it unclear what practical insights the theory provides beyond general intuition that adaptive filtering could be beneficial.

**W3**. Comparing Table 1 with the original GADBench paper [4] reveals substantial discrepancies that are difficult to reconcile. While some variation is expected across different hardware/software configurations, differences of this magnitude are concerning, particularly since the authors claim to use GADBench's implementation and evaluation protocol.

**W4**. The paper exclusively reports AUPRC, AUROC, and Rec@K, but omits F1 score—a standard and important metric for imbalanced classification that provides complementary information about precision-recall tradeoffs.

---

**Reference**

[1] J. Tang et al. *Rethinking Graph Neural Networks for Anomaly Detection*. ICML 2022.

[2] Y. Gao et al. *Addressing Heterophily in Graph Anomaly Detection: A Perspective of Graph Spectrum*. WWW 2023.

[3] X. Dong et al. *Rayleigh Quotient Graph Neural Networks for Graph-level Anomaly Detection*. ICLR 2024

[4] J. Tang et al. *GADBench: Revisiting and Benchmarking Supervised Graph Anomaly Detection*. NeurIPS 2023.

**Questions:**

**Q1**.What are the substantial discrepancies between your reported baseline results and those in the original GADBench paper? Please provide a detailed explanation.

**Q2**. F1 scores for all methods in the main experimental results should also be reported.

**Q3**. How does the theoretical analysis in Theorem 1 relate to the actual deployment of APF?

**Details Of Ethics Concerns:**

No concerns

---

> ### Author Response · Authors · 2025-11-19
> **Official Response by Authors (1)**
>
> Thanks for your valuable suggestions. We will try to address your concerns, and we are eager to engage in a more detailed discussion with you.
>
> > W1: Limited Technical Novelty
>
> Thank you for this critical and insightful comment. It allows us to clarify the primary motivation and technical novelty of our work.
>
> We agree that the individual components, such as the Rayleigh Quotient and spectral filtering, are well-established techniques. However, we would like to respectfully clarify that our primary contribution is not the invention of these components, but rather the identification and principled solution of a **critical research problem**: *How to design a GAD-specific pre-training and fine-tuning framework to unlock its full potential for label-scarce anomaly detection.*
>
> Our starting point was the observation that general-purpose pre-training models can surprisingly outperform even GAD-specific supervised models under label scarcity. This suggested that while the *paradigm* is promising, its full potential for GAD remains untapped because existing frameworks are task-agnostic.
>
> This motivated us to address the key gaps we identified: existing pre-training paradigms lack (1) an **anomaly-aware pre-training objective** and (2) a **corresponding backbone architecture** designed to capture both semantic and anomaly-specific information.
>
> The core of our technical contribution is *how* we fill these gaps. Rather than offering a trivial mix of techniques, our design is a principled framework where each component follows as the natural means to bridge these paradigm gaps:
>
> 1. **An Anomaly-Aware Pre-Training Objective:** While the Rayleigh Quotient has been used in *supervised* GAD, its adaptation as a **label-free anomaly indicator to guide an *unsupervised* pre-training objective** is a key part of our novelty. We use it to create our anomaly-aware objective, which explicitly trains the high-pass filter ($Z_H$) to be sensitive to these structural deviations *before any labels are seen*. This is a clear departure from task-agnostic pre-training.
> 2. **A Necessary Backbone:** The **Dual-filter Encoding** is the necessary architectural counterpart to this dual-objective. This backbone is *required* to simultaneously learn and *decouple* general semantics ($Z_L$) from the anomaly-aware cues ($Z_H$). A standard low-pass-only backbone, as used in general pre-training, would be unable to capture the high-frequency signals our new objective is designed to find.
> 3. **A Tailored Fine-Tuning Stage:** This pre-training design enables our *second* key technical innovation: the **granularity-adaptive fine-tuning stage**. This stage, featuring a **gated fusion network** and an **anomaly-aware regularization loss**, is specifically designed to *maximize the utility* of the decoupled representations ($Z_L$ and $Z_H$) learned during pre-training. This fusion mechanism is what allows the model to adaptively handle the homophily disparity we identified as a key GAD challenge.
>
> In summary, APF fills these research gaps through its principled two-stage design: (1) pre-training with a novel, GAD-specific objective to learn decoupled representations, and (2) fine-tuning with an adaptive fusion network to best exploit them.

---

> ### Author Response · Authors · 2025-11-19
> **Official Response by Authors (2)**
>
> > W2&Q3: Theory–Method Mismatch Reducing Practical Relevance
>
> Thank you for carefully reading Theorem 1. We clarify how the theorem relates to the deployed APF below:
>
> **(i) Gaussian feature assumption in ASBM**
>
> We agree that real-world node features are often heterogeneous, sparse, or even categorical. In our theory, we adopt Gaussian contextual features primarily **to make the analysis tractable and derive closed-form separability conditions**, following a long line of work on GNN/GAD theory. In particular, assuming Gaussian node features is standard in theoretical analyses of GNNs and GAD, and has been shown to provide useful guidance for architecture design, e.g.:
>
> * GNN separability under contextual SBM with Gaussian features [1];
> * GAD under similar modeling assumptions [2].
>
> Our use of the ASBM is in the same spirit: it is not intended as an exact generative model for all GADBench datasets, but as a simplified sandbox to isolate the effect of **node-adaptive low-/high-pass filtering under homophily disparity**.
>
> Empirically, we also observe that APF works well on datasets whose node features are **far from Gaussian**, such as YelpChi and T-Social, which contain highly heterogeneous tabular attributes. This suggests that the architectural insight derived from the Gaussian setting is robust beyond this idealized assumption, even though the theorem itself is proved under ASBM.
>
> [1] *Graph Convolution for Semi-Supervised Classification: Improved Linear Separability and Out-of-Distribution Generalization.* ICML 2021
>
> [2] *Rethinking Graph Neural Networks for Anomaly Detection.* ICML 2022
>
> ---
>
> **(ii) Oracle homophily patterns vs data-driven gated fusion**
>
> Theorem 1 analyzes an *idealized* scenario where we know, for each node, whether it is homophilic or heterophilic and apply low-pass and high-pass filters accordingly. In practice, APF does **not** use such discrete pattern assignments.
>
> We aim to show that, *if* one could assign the "right" filter to each node based on its local homophily, then the resulting filtered features are linearly separable with high probability. This provides a target behavior for the architecture to approximate.
>
> In APF, this oracle assignment is replaced by the **gated fusion network (GFN)** and **anomaly-aware regularization**:
>
> * We always compute *both* filtered representations $Z_L$ and $Z_H$;
> * The GFN generates continuous fusing coefficients $C$, and the fused representation is a *soft, learnable relaxation* of the hard "low-pass vs high-pass" assignment in Theorem 1;
> * The regularization explicitly encourages class-dependent fusion (abnormal nodes relying more on the anomaly-sensitive branch, normal nodes more on the semantic branch), which is consistent with the homophily disparity statistics we report.
>
> Thus, while the theorem assumes perfect knowledge of homophily patterns, **the architecture is designed precisely to approximate this ideal allocation in a data-driven way**, and our visualizations of $C$ (Fig. 4) and ablations (Table 3) empirically support that the learned gates correlate with structural disparity.
>
> ---
>
> **(iii) Linear classifier on frozen features vs full APF**
>
> Theorem 1 is proved for a **linear classifier on frozen, filtered features**, whereas the deployed APF includes learnable polynomial filters, MLP encoders, and a trainable fusion and classifier.
>
> This simplification is as the goal of Theorem 1 to isolate and analyze the **core mechanism** of APF: *node-specific combination of low-pass and high-pass information under homophily disparity*, rather than provide a complete optimization guarantee for the full deep network. Concretely:
>
> * The theorem says that **there exists a linear separator** once the ideal node-wise filtering is applied.
> * The APF architecture can be viewed as a *parameterized implementation* that (i) learns the filters, (ii) learns a soft assignment between them via GFN, and (iii) uses an MLP classifier that is at least as expressive as the linear classifier in the theorem.
> * Our ablation study (Table 3) shows that when we remove dual filters, Rayleigh Quotient guidance, or adaptive fusion, performance drops significantly, which empirically supports the theoretical insight that **adaptive dual filtering is the key factor**.
>
> ---
>
> **Summary: Practical insight of Theorem 1 for APF**
>
> In summary, Theorem 1:
>
> * is built on standard simplifying assumptions to obtain a **clean characterization of linear separability under node-adaptive filtering**;
> * directly motivates APF's architecture: *dual spectral filters* plus *learned, node-wise fusion* that responds to homophily disparity;
> * is complemented by extensive experiments and ablations showing that these theoretically motivated components are indeed crucial in practice, including on heterogeneous, non-Gaussian datasets.
>
> **ACTION**: We have revised Section 3.3 Theoretical Insights and added Appendix J to clarify the theorem better. Thank you again for your valuable feedback.

---

> ### Author Response · Authors · 2025-11-19
> **Official Response by Authors (3)**
>
> > W3&Q1: Large Discrepancies from GADBench Baseline Results
>
> Thank you for your careful review and for raising this point. We appreciate the opportunity to clarify this discrepancy.
>
> The GADBench paper introduces two distinct experimental settings:
>
> 1. A **Fully-supervised** setting, which uses a large portion of the nodes for training (e.g., 40% for training, 20% for validation, as described in their main paper).
> 2. A **Semi-supervised** setting, which is designed to specifically mimic real-world label scarcity.
>
> As the central motivation of our paper is to address GAD under label scarcity, we strictly and exclusively follow the semi-supervised setting defined by GADBench. This protocol standardizes the training set across all datasets to include only 100 total labels (20 positive/anomalous nodes and 80 negative/normal nodes) and uses the 10 data splits provided by GADBench for robust evaluation.
>
> The baseline results in our Table 1 are taken directly from **Table 11 in the Appendix of the GADBench paper**, which explicitly reports the performance for this semi-supervised setting.
>
> We believe the discrepancy you noted arises from comparing our Table 1 (semi-supervised) with the main results tables in the GADBench paper (e.g., their Table 4), which report on the **fully-supervised** setting. The performance numbers from these two different settings are, by design, not comparable.
>
> For any baselines in our paper that were not included in GADBench's Table 11, we used the authors' official code and evaluated them under the exact same semi-supervised protocol.
>
> We can therefore confirm that our experimental setup and reported baseline results are fully consistent with the GADBench semi-supervised protocol.
>
> **ACTION**: We have added a clarification to Section 4.1 Experimental Setup and Appendix H.3 Evacuation Protocols to explicitly state that our comparisons refer to the semi-supervised setting of GADBench to prevent future confusion. Thank you again for your valuable feedback.
>
>
> ---
>
>
>
> > W4&Q2: Missing F1 Metric
>
> Thank you for the helpful suggestion. Our choice of evaluation metrics (AUPRC, AUROC, and Rec@K) was made to strictly follow the established protocol of GADBench, which is the benchmark framework we used for our 10 datasets and experimental setup.
>
> We agree that the F1 score is a standard metric for imbalanced classification. However, we believe the combination of AUPRC and Rec@K provides a comprehensive and arguably more robust assessment of the precision-recall tradeoffs for the following reasons:
>
> 1. **AUPRC vs. F1 Score:** The F1 score is a *threshold-dependent* metric. To report a single F1 score, we would need to select an "optimal" decision threshold based on the validation set. In our label-scarce setting (with only 100 labeled nodes, 20 of which are anomalies), this threshold selection can be noisy and unstable. In contrast, the AUPRC (Area Under the Precision-Recall Curve) is a threshold-independent metric that summarizes the model's precision-recall tradeoff across all possible thresholds, providing a more complete and robust measure of performance.
> 2. **Practical Relevance of Rec@K:** The Rec@K metric directly evaluates the practical utility of a GAD system. In real-world applications (e.g., fraud detection), an analyst typically reviews only the "top-K" most suspicious items. Rec@K measures the recall within this top-K list (where K is the total number of anomalies in the test set), directly assessing the model's crucial ability to rank true anomalies highly.
>
> Given that our chosen metrics adhere to the standard GAD benchmark and provide a robust, threshold-independent (AUPRC, AUROC) and practical (Rec@K) assessment, we believe they offer a comprehensive evaluation of our model's performance.
>
> **ACTION**: We have added a clarification to Section 4.1 Experimental Setup and Appendix H.3 Evacuation Protocols to explain why we adopt AUPRC, AUROC, and Rec@K against F1 score. Thank you for guiding us to strengthen this aspect of our paper.

---

> > ### Comment · Reviewer_sLnc · 2025-11-27
> > **Comment**
> >
> > I thank the authors for their thorough response. Most of my concerns have been addressed. Although I still stand by my opinion, as the rationale for connecting the theoretical analysis under ideal settings with the empirical findings remains doubtful, I will raise my initial score to encourage the authors' future exploration in this direction.

---

> > > ### Author Response · Authors · 2025-11-27
> > > **Thank you again!**
> > >
> > > Thank you sincerely for your follow-up comment and for kindly raising your score. We truly appreciate the time and thought you have invested in reviewing our work.
> > >
> > > We fully understand your remaining concerns regarding the gap between the idealized theoretical setup and the practical APF framework. Your feedback has helped us clarify the intended scope of the analysis and also highlighted meaningful directions for future work. Bridging theory with realistic, adaptive GNN architectures is indeed challenging, and your comments motivate us to further strengthen this aspect in our subsequent research.
> > >
> > > Thank you again for your constructive insights and supportive attitude. We are grateful for your engagement throughout the discussion.

---

### Official Review · Reviewer_HGx6 · 2025-11-02

**Soundness:** 3
**Presentation:** 3
**Contribution:** 2
**Rating:** 6
**Confidence:** 4

**Summary:**

This paper introduces AFP, a GAD framework designed to address label scarcity and local homophily disparity through a two-stage design. In the pre-training stage, a Rayleigh Quotient and dual spectral filter-based module are trained to learn anomaly-sensitive information. In the fine-tuning stage, a fusion network and anomaly-aware regularization adapt to the test graph. Theoretical analyses are provided for separability. The proposed method shows competitive performance on the selected benchmark datasets. My view is that the proposed method is an improved strategy to take the advantages of both unsupervised and supervised training.

**Strengths:**

1. The proposed method and problem setting are interesting and well-motivated.
2. The paper provides theoretical analyses to justify the model design.
3. A comprehensive set of benchmark datasets is used, and the method achieves competitive performance across them.

**Weaknesses:**

1. While many baseline methods are included, they should be categorized by the type of anomaly information used (i.e., supervised, semi-supervised, or unsupervised) to clarify performance differences.
2. The paper explores a unique within-graph pre-training and fine-tuning setting. A discussion comparing it with other GAD settings using label information, such as meta-learning GAD (Meta-GDN, Ding et al., 2021), cross-domain GAD (Commander, Ding et al., 2021; ACT, Wang et al., 2023), and the more recent spectral paper (DSGAD, Zheng et al., 2025) would better position the contribution.
3. Most baselines are purely unsupervised or purely supervised. It would be helpful to discuss how combining supervised and unsupervised strategies (joint learning) would perform compared to the proposed method.
4. Since pre-training generally enables faster adaptation under distribution shift, could the authors comment on whether the proposed approach could extend to cross-dataset GAD or detecting novel types of anomalies as studied in (PreNet, Pang et al., 2022; GDN-AugAN, Zhou et al., 2023; and NSReg, Wang et al., 2025)?

**Questions:**

Please refer to my weaknesses.

---

> ### Author Response · Authors · 2025-11-19
> **Official Response by Authors (1)**
>
> Thanks for your valuable suggestions. We will try to address your concerns, and we are eager to engage in a more detailed discussion with you.
>
> > W1: Baseline Categorization Issue
>
> Thank you for this very constructive suggestion.
>
> Our current categorization in the paper was designed to distinguish models by their architectural purpose: (1) Standard GNNs (general-purpose models not designed for GAD), (2) Specialized GNNs (models designed specifically for GAD), and (3) Pre-training/Fine-tuning (architectures designed to learning better representations for downstream tasks).
>
> As you've pointed out, re-categorizing the baselines based on the type of anomaly information used provides a clearer perspective on performance differences, especially under the label-scarce settings we address. Following your advice, we can re-classify the baselines as follows:
>
> - **Supervised** (models directly trained with labeled anomalies): GCN, GIN, GAT, ACM, FAGCN, AdaGNN, BernNet, GAS, PCGNN, AMNet, BWGNN, GHRN.
> - **Semi-supervised** (models that can effectively leverage limited anomaly labels): ConsisGAD, SpaceGNN, XGBGraph. (Note that these are essentially *a subclass of supervised methods* but are specialized for limited-label scenarios).
> - **Self-supervised** (models first pre-trained without any labels, then fine-tuned with labeled anomalies): DCI, DGI, GRACE, G-BT, GraphMAE, BGRL, SSGE, PolyGCL, BWDGI, and our **APF**.
>
> To highlight the insights from this new grouping, we provide a table comparing the top-3 average AUPRC performers from each category, along with our proposed APF.
>
> |Category|Model|Avg. AUPRC|Avg. AUROC|Avg. Rec@K|
> |-|-|-|-|-|
> |**Supervised**|PCGNN|32.9|73.8|34.6|
> ||BWGNN|35.4|76.0|37.4|
> ||GHRN|35.4|76.1|37.7|
> |**Semi-supervised**|ConsisGAD|38.6|78.0|39.6|
> ||SpaceGNN|41.8|79.0|42.2|
> ||XGBGraph|42.9|77.8|41.3|
> |**Self-supervised**|SSGE|37.1|77.6|38.2|
> ||PolyGCL|37.1|78.7|38.6|
> ||BWDGI|39.2|78.2|39.2|
> ||**APF (Ours)**|**49.6**|**82.8**|**48.4**|
>
> This new categorization reveals two key findings that strongly support our paper's motivation:
>
> 1. In this realistic label-scarce scenario, **general-purpose self-supervised models** already outperform the **best-specialized supervised GAD models** (e.g., GHRN/BWGNN) by 2-4 percentage points in average AUPRC. This demonstrates the significant potential of the pre-training paradigm for handling the label scarcity inherent in GAD.
> 2. Our **APF**, which is a self-supervised model specifically optimized for GAD, achieves an average AUPRC of 49.6. This markedly outperforms the strongest **semi-supervised GAD models** (e.g., SpaceGNN at 41.8) by nearly 8 percentage points and the best **general self-supervised models** (e.g., BWDGI at 39.2) by over 10 percentage points.
>
> This analysis strongly validates our core motivation: a self-supervised (pre-training then fine-tuning) framework *specifically tailored* for the challenges of GAD is crucial and effective.
>
> **ACTION**: We have included this detailed breakdown and analysis to Appendix I.8 Performance Analysis by Supervision Paradigm in the revised manuscript. Thank you again for your valuable feedback.

---

> ### Author Response · Authors · 2025-11-19
> **Official Response by Authors (2)**
>
> > W2: Insufficient Positioning Against Other Label-Utilizing GAD Settings
>
> We sincerely thank the reviewer for these valuable references. We agree that discussing these works will significantly clarify our paper's position within the broader GAD landscape.
>
> Our proposed **APF** focuses on a **"Single-Graph Pre-training and Fine-tuning"** setting. The core challenge we address is how to effectively detect anomalies in a *single* target graph where labels are extremely scarce, *without* relying on any external auxiliary networks or source domains.
>
> We distinguish our work from the suggested settings as follows:
>
> **1. vs. Cross-Network / Meta-Learning GAD (e.g., Meta-GDN [1])**
>
> - **Setting Difference:** Meta-learning approaches like Meta-GDN assume the availability of **multiple auxiliary networks** from the same domain to learn transferable meta-knowledge for few-shot detection on the target graph.
> - **Our Position:** In many real-world scenarios, collecting high-quality auxiliary networks is difficult or impossible (e.g., a unique internal financial transaction network). APF is designed to be **self-reliant**; it does not require any auxiliary graphs. Instead, it fully exploits the abundant *unlabeled* nodes within the *same* graph via anomaly-aware pre-training to initialize the model, enabling effective fine-tuning with limited labels.
>
> **2. vs. Cross-Domain GAD (e.g., Commander [2], ACT [3])**
>
> - **Setting Difference:** Cross-domain methods assume the existence of a **labeled source domain graph** and focus on addressing the domain shift to transfer knowledge to an unlabeled (or few-shot) target graph.
> - **Our Position:** APF operates in a **within-domain** setting. We avoid the challenges of domain adaptation (e.g., negative transfer due to large distribution shifts) and the costs of acquiring source domain data. APF focuses on mining intrinsic anomaly signals (via the Rayleigh Quotient and spectral filters) directly from the target graph's own structure, making it applicable when no related source domain exists.
>
> **3. vs. Dynamic Spectral GAD (e.g., DSGAD [4])**
>
> - **Methodological Difference:** DSGAD is a recent extension of the spectralized GAD model BWGNN [5], improving it with trainable dynamic wavelets and a more consistent feature fusion mechanism. It primarily focuses on **architectural innovations** (e.g., dynamic wavelets, feature fusion) trained in an **end-to-end semi-supervised** manner.
>
> - **Our Position:** APF introduces a **learning paradigm shift** (Pre-training then Fine-tuning) rather than just a new GNN architecture. While we also utilize spectral filters, we decoupe the learning process:
>
>   1. **Pre-training:** Unsupervised capturing of general semantics and anomaly-aware cues (via dual-filter encoding).
>   2. **Fine-tuning:** Adaptive fusion tailored to label scarcity and homophily disparity.
>
>   - As shown in our experiments (Table 1), this pre-training paradigm allows APF to significantly outperform end-to-end spectral models (like BWGNN/GHRN) in label-scarce settings by providing a more robust initialization that prevents overfitting to few labels.
>
> **Summary of Contribution:**
>
> Our work fills a critical gap: How to maximize detection performance on a single graph with scarce labels when no external data (auxiliary networks or source domains) is available.
>
> **ACTION**: We have added a discussion of these works in Appendix C Related Works to better position APF. Thank you for guiding us to strengthen this aspect of our paper.
>
> [1] Ding et al. *Few-shot Network Anomaly Detection via Cross-network Meta-learning*. WWW 2021
>
> [2] Ding et al. *Cross-Domain Graph Anomaly Detection*. TNNLS 2021
>
> [3] Wang et al. *Cross-Domain Graph Anomaly Detection via Anomaly-aware Contrastive Alignment*. AAAI 2023
>
> [4] Zheng et al. *Dynamic Spectral Graph Anomaly Detection*. AAAI 2025
>
> [5] Tang et al. *Rethinking Graph Neural Networks for Anomaly Detection*. ICML 2022

---

> ### Author Response · Authors · 2025-11-19
> **Official Response by Authors (3)**
>
> > W3: Lack of Discussion on Joint Supervised + Unsupervised Strategies
>
> Thank you for this valuable suggestion. It is an important comparison to justify our two-stage (pre-training then fine-tuning) design choice.
>
> Following your recommendation, we conducted an experiment to compare our proposed method against a **joint learning** strategy. We define the two strategies as:
>
> - **Two-stage (Ours):** First, the model is trained with the unsupervised pre-training objective, and then this model is fine-tuned using the supervised binary classification loss.
> - **Joint Learning:** The model is trained end-to-end by simultaneously optimizing supervised loss and unsupervised loss.
>
> We applied these strategies to the GCN and BWGNN baselines (using DGI as the pre-training objective) as well as to our APF framework. The AUPRC results on four datasets are presented in the table below:
>
> |Model|Training Strategy|Reddit|YelpChi|Tolokers|DGraph-Fin|
> |-|-|-|-|-|-|
> |GCN|Supervised-only|4.2±0.8|16.4±2.6|33.0±3.6|2.3±0.2|
> |GCN+DGI|Two-stage|4.8±0.6|17.0±1.2|39.7±0.8|2.1±0.2|
> |GCN+DGI|Joint learning|4.4±1.0|18.2±2.3|38.0±3.4|2.3±1.4|
> |||||||
> |BWGNN|Supervised-only|4.2±0.7|23.7±2.9|35.3±2.2|2.1±0.3|
> |BWGNN+DGI|Two-stage|4.5±0.6|26.8±2.7|38.5±3.1|2.4±0.2|
> |BWGNN+DGI|Joint learning|4.3±0.6|26.8±4.1|38.2±3.4|2.3±0.2|
> |||||||
> |APF|Supervised-only|5.2±0.6|24.1±2.2|37.4±1.2|2.3±0.2|
> |**APF**|**Two-stage (Ours)**|**5.9±0.9**|**28.4±1.4**|**40.5±2.0**|**2.9±0.2**|
> |APF|Joint learning|5.5±0.7|28.0±1.8|39.6±2.2|2.7±0.3|
>
> From these results, we can draw two key conclusions:
>
> 1. **Joint learning is beneficial:** As the reviewer anticipated, combining supervised and unsupervised strategies in a joint learning framework generally improves performance over purely supervised methods (e.g., GCN+DGI joint learning vs. GCN).
> 2. **Two-stage is superior:** In almost all cases, the two-stage pre-training then fine-tuning paradigm performs better than the joint learning strategy. This is especially clear for our proposed APF, which achieves the best performance with its two-stage design.
>
> We hypothesize this is because the goals of the unsupervised pre-training objective and the supervised classification loss can conflict when optimized simultaneously, leading to suboptimal representations.
>
> This finding is also consistent with existing literature. For instance, Wang et al. (2021) [1] similarly found that decoupling representation learning from classification (i.e., a two-stage approach) was more effective than joint training for GNN-based anomaly detection. This experiment further validates our design choice of a decoupled, two-stage framework for this task.
>
> **ACTION**: We have included this discussion and the results table in Appendix I.9 Comparison with Joint Learning Strategy to clarify the relative advantages of joint learning versus two-stage pre-training. Thank you again for your valuable feedback.
>
> [1] Wang et al. *Decoupling Representation Learning and Classification for GNN-based Anomaly Detection*. SIGIR 2021

---

> ### Author Response · Authors · 2025-11-19
> **Official Response by Authors (4)**
>
> > W4: Unclear Applicability to Distribution Shift or Novel Anomaly Types
>
> Thank you for this insightful point. We agree that pre-training often facilitates adaptation under distribution shift, and the question of whether APF can extend to cross-dataset GAD or unseen-anomaly detection (as studied in PReNet [1], GDN-AugAN [2], NSReg [3]) is important. Below we discuss the potential extensions in two parts:
>
> **1. Detecting Novel/Unseen Anomaly Types (Open-Set GAD)**
>
> This setting, explored by PReNet [1] and NSReg [3], aims to detect new types of anomalies that were not present in the small labeled training set.
>
> - **The Challenge:** Standard semi-supervised models risk "overfitting" to the specific patterns of the *seen* anomalies. As noted in NSReg, this can cause the model to misclassify *unseen* anomalies (with different patterns) as normal.
> - **How APF Could Help:** APF is uniquely positioned to mitigate this. Our pre-training stage is entirely **unsupervised** and does not rely on any specific anomaly examples.
>   - It learns two fundamental and complementary representations: a low-pass representation ($Z_L$) capturing **general semantic patterns** (i.e., a strong model of "normality") and a high-pass representation ($Z_H$) capturing **subtle anomaly cues** (i.e., a general model of "deviation").
>   - Crucially, the anomaly-aware branch is guided by the **Rayleigh Quotient**, a label-free metric for structural diversity and potential anomaly signals.
> - **Our Hypothesis:** Because APF pre-trains a strong, general-purpose model of both "normality" ($Z_L$) and "label-agnostic deviation" ($Z_H$), the fine-tuning stage is less about learning anomaly patterns from scratch and more about learning *how to best fuse* these pre-learned concepts. This approach is inherently more general than an end-to-end model that only learns to separate the seen anomalies from normal nodes. This design closely aligns with the goal of NSReg, which enforces a "Normal Structure Regularisation" to learn a better normality decision boundary. APF achieves this by default through its pre-trained low-pass filter.
>
> **2. Extending to Cross-Dataset GAD**
>
> This setting, explored by GDN-AugAN [2], involves generalizing a trained model to new, unseen graphs or subgraphs.
>
> - **The Challenge:** Models trained on one graph often fail to generalize to another due to distribution shifts in both features and graph structure.
> - **How APF Could Help:** This is a classic use case for pre-training. The encoders learned during APF's **anomaly-aware pre-training** (the dual spectral polynomial filters) could be used as a powerful, transferable feature extractor.
> - **Our Hypothesis:** One could pre-train the APF encoders on a large-scale graph (or multiple graphs) and then transfer this pre-trained model to a new target dataset, followed by fine-tuning on the few available labels from that new graph. Because our pre-training is not just task-agnostic but anomaly-aware, it would provide a much more relevant initialization for a new GAD task, likely leading to faster adaptation and superior performance compared to training from scratch or using generic pre-trained models.
>
> In summary, this is a very promising direction for future work. The fundamental design of APF, decoupling the learning of general normality/deviation (pre-training) from task-specific adaptation (fine-tuning), is precisely what is needed to build more generalizable GAD models.
>
> **ACTION**: We have added a discussion of these works in Appendix C Related Works. Thank you again for this insightful suggestion.
>
> [1] Pang et al. *Deep Weakly-supervised Anomaly Detection*. KDD 2023
>
> [2] Zhou et al. *Improving Generalizability of Graph Anomaly Detection Models via Data Augmentation*. TKDE 2023
>
> [3] Wang et al. *Open-Set Graph Anomaly Detection via Normal Structure Regularisation*. ICLR 2025

---

> > ### Comment · Reviewer_HGx6 · 2025-11-27
> >
> > Many thanks to the authors for your thorough response. My concerns are mostly addressed. I will therefore maintain my initial rating in support of this paper.

---

> > > ### Author Response · Authors · 2025-11-27
> > > **Thank you again!**
> > >
> > > Thank you for your thoughtful comments and for revisiting our detailed response. We are pleased that our clarifications addressed your main concerns, and we truly appreciate your supportive assessment of the paper.
> > >
> > > Your suggestions have significantly improved the clarity and positioning of our work, and we are grateful for your time and insight. If any additional thoughts arise during the remaining discussion period, we would be glad to continue the conversation.
> > >
> > > Thank you again for your constructive feedback.

---

### Author Response · Authors · 2025-11-27
**Summary of Revisions and Responses to Reviewers**

Dear Reviewers and Area Chair,

We sincerely thank you for the thoughtful and constructive feedback, which has been invaluable in improving the clarity, positioning, and empirical strength of our work. We have uploaded a revised manuscript incorporating the changes made during the rebuttal.

We understand that our esteemed reviewers have a multitude of responsibilities. To facilitate a quick understanding, we have summarized the key points of the responses for your convenience:

**1. Clarified Technical Novelty & Contribution** *(for sLnc, TPKe)*

We have revised the methodology and appendix to clarify that our contribution goes beyond using established components, but the principled design of an anomaly-aware pre-training framework. We highlight how the Dual-filter Encoding and Anomaly-Aware Objective are co-designed to explicitly solve the *label scarcity* and *homophily disparity* problems, a gap that task-agnostic pre-training cannot address.

**2. Strengthened Empirical Validation** *(for HGx6, sLnc)*

- **Baseline Re‑categorization:** Grouped baselines by supervision paradigm (Supervised / Semi‑supervised / Self‑supervised), showing APF clearly outperforms even the strongest semi‑supervised GAD models under label‑scarce conditions.
- **Joint vs. Two‑Stage Learning:** Added experiments (Appendix I.9) demonstrating that our decoupled pre‑training → fine‑tuning paradigm consistently outperforms joint supervised+unsupervised optimization.
- **Protocol Clarification:** Explicitly confirmed strict adherence to the *semi‑supervised GADBench protocol* (20 labeled anomalies), resolving discrepancies with fully‑supervised results.

**3. Improved Theoretical and Related‑Work Positioning** *(for EFtV, HGx6)*

- **Related Works:** Expanded Appendix C to distinguish APF from concurrent and related works, e.g., UniGAD’s use of RQ for multi‑task format unification vs. our use for single‑graph anomaly‑aware pre-training.
- **Theory–Method Link:** Revised Sec. 3.3 and added Appendix J to clarify how Theorem 1 (under ASBM) motivates our architectural choice of gated fusion network for adaptive low/high‑pass integration.

We hope these revisions address the main concerns raised and more clearly convey the practical value of APF. We would be very grateful for any further comments during the remaining discussion time, should you wish to share them.

Thank you again for your time and consideration.

Best regards,
The Authors

---

### Author Response · Authors · 2025-12-01
**Summary of Review Timeline for Area Chair**

Dear new Area Chair,

We thank you and the original AC/reviewers for your service and hard work on our submission. We understand the difficult circumstances and appreciate you stepping into this critical role.

To assist you in your decision-making process, we would like to provide a concise summary of our paper's review timeline. All discussions and score changes for our paper occurred *before* the public disclosure of the OpenReview security incident.

**Review Timeline:**

- **Nov 12, 00:22 AoE:** Initial reviews were released. Our paper received scores of **6, 6, 2, 6**.

- **Nov 18, 23:15–23:35 AoE:** We uploaded our revised paper and posted detailed responses to all reviewers.

- **Nov 26, 14:29 AoE:** We posted a summary of our revisions to encourage further discussion.

- **Nov 26, 14:49–15:25 AoE:** Reviewers EFtV, TPKe, and HGx6 acknowledged our response and maintained their positive scores (6).

- **Nov 26, 19:18 AoE:** Following our rebuttal, Reviewer sLnc raised their score from **2 to 6** (please kindly see the revision history of Reviewer sLnc's official review), providing the following comment:

  > "I thank the authors for their thorough response. Most of my concerns have been addressed. Although I still stand by my opinion, as the rationale for connecting the theoretical analysis under ideal settings with the empirical findings remains doubtful, I will raise my initial score to encourage the authors' future exploration in this direction."

- **~Nov 27, 03:00–04:10 AoE:** The OpenReview API security incident occurred and was addressed (per the official "Statement Regarding API Security Incident" from OpenReview News).


Despite the subsequent rollback of reviews to their pre-discussion state, we hope the above information provides useful context. All of our responses remain in the discussion and reflect our genuine efforts to address reviewer concerns thoroughly and constructively. We appreciate your time and consideration, and we are grateful for the community's collective effort in navigating this unexpected challenge.

Thank you again for your time and careful consideration.

Sincerely,
The Authors

---

### Meta-Review · Area_Chair_SzHU · 2026-01-13

**Summary:**

The paper proposes "Anomaly-Aware Pre-Training and Fine-Tuning" (APF), a framework designed to address Graph Anomaly Detection (GAD) under conditions of label scarcity and homophily disparity. The method introduces a pre-training stage utilizing the Rayleigh Quotient and dual spectral filters to capture anomaly-aware signals without labels, followed by a granularity-adaptive fine-tuning stage.
The reviewers unanimously appreciated the clear motivation and the practical relevance of the problem setting (label scarcity). The extensive experimental evaluation across 10 benchmark datasets was cited as a major strength. Initial concerns focused primarily on the technical novelty (perceived as a combination of existing components), the alignment between the theoretical analysis and the practical implementation, and specific experimental details regarding baseline comparisons.

**Reviewer Concerns:**

Addressed Concerns:
1. Novelty and Contribution (Reviewers sLnc, TPKe): Reviewer sLnc initially questioned whether the combination of Rayleigh Quotient sampling and spectral filtering was sufficiently novel. The authors successfully clarified that the contribution lies in the principled framework design specifically tailored for GAD, rather than the components themselves. They argued effectively that general-purpose pre-training fails to capture anomaly cues, necessitating their specific design.
2. Experimental Discrepancies and Baselines (Reviewers sLnc, HGx6): Reviewer sLnc noted discrepancies between reported results and the original GADBench paper. The authors clarified that they strictly followed the semi-supervised protocol (100 labels) rather than the fully supervised setting, resolving the confusion. Furthermore, the authors addressed HGx6's request by re-categorizing baselines (Supervised vs. Semi-supervised vs. Self-supervised), which clearly demonstrated APF's superiority in label-scarce regimes.
3. Comparison with Joint Learning (Reviewer HGx6): The authors added experiments (Appendix 1.9) comparing their two-stage approach with a joint learning strategy, empirically proving the superiority of the proposed decoupled framework.

Outstanding Concerns:
1. Theoretical vs. Practical Gap (Reviewer sLnc): While Reviewer sLnc raised their score to accept, they maintained a reservation regarding Theorem 1. The theoretical analysis relies on idealized assumptions (Gaussian features, oracle knowledge of homophily patterns) which differ from the actual data-driven Gated Fusion Network used in APF. The reviewer accepted the authors' explanation that the theory serves as a motivational "sandbox," but noted the connection remains "doubtful." This is a valid limitation but not a rejection factor.

**Reviewer Scores:**

The reviewers actively participated in the discussion, and the scores reflect their final positions.
1. Reviewer sLnc: Raised score from 2 to 6 (possibly) after the rebuttal addressed concerns about novelty and experimental protocols.
2. Reviewer HGx6: Maintained a score of 6, noting that their concerns were "mostly addressed" and the baseline categorization improved the paper.
3. Reviewer TPKe: Maintained a score of 6.
4. Reviewer EFtV: Maintained a score of 6.

---

### Decision · Program_Chairs · 2026-01-26

Accept (Poster)